



# Aitken mode particles as CCN in aerosol- and updraft-sensitive regimes of cloud droplet formation

Mira L. Pöhlker[1], Minghui Zhang[2], Ramon Campos Braga[1], Ovid O. Krüger[1], Ulrich Pöschl[1], and Barbara Ervens[2]

[1]Multiphase Chemistry Department, Max Planck Institute for Chemistry, 55128 Mainz, Germany
[2]Université Clermont Auvergne, CNRS, SIGMA Clermont, Institut de Chimie de Clermont-Ferrand, 63000 Clermont-Ferrand, France

**Correspondence:** Mira Pöhlker (m.pohlker@mpic.de), Barbara Ervens (barbara.ervens@uca.fr)

**Abstract.** The high variability of aerosol particle concentrations, sizes and chemical composition makes their description challenging in atmospheric models. Aerosol-cloud interaction studies are usually focused on the activation of accumulation mode particles as cloud condensation nuclei (CCN). However, under specific conditions also Aitken mode particles can contribute to the number concentration of cloud droplets ($N_d$), leading to large uncertainties in predicted cloud properties on a global scale.

We perform sensitivity studies with an adiabatic cloud parcel model to constrain conditions, under which Aitken mode particles contribute to $N_d$. The simulations cover wide ranges of aerosol properties, such as total particle number concentration, hygroscopicity ($\kappa$) and mode diameters for accumulation and Aitken mode particles. Building upon the previously suggested concept of updraft ($w$)- and aerosol-limited regimes of cloud droplet formation, we show that activation of Aitken mode particles does not occur in $w$-limited regimes of accumulation mode particles. The transitional range between the regimes is broadened when

Aitken mode particles contribute to $N_d$ as aerosol-limitation requires much higher $w$ than for aerosol size distributions with accumulation mode particles only. In the transitional regime, $N_d$ is similarly dependent on $w$ and $\kappa$. Therefore, we analyze the sensitivity of $N_d$ to $\kappa$, $\xi(\kappa)$, as a function of $w$ to identify the value combinations, above which Aitken mode particles can affect $N_d$. As $\xi(\kappa)$ shows a minimum when the smallest activated particle size is in the range of the 'Hoppel minimum' (0.06 μm $\leq D_{min} \leq 0.08$ μm), the corresponding ($w,\kappa$) pairs can be considered a threshold level, above which Aitken mode particles

have significant impact on $N_d$. This threshold is largely determined by the number concentration of accumulation mode particles and by the Aitken mode diameter. Our analysis of these thresholds results in a simple parametric framework and criterion to identify aerosol and updraft conditions, under which Aitken mode particles are expected to affect aerosol-cloud interactions. Our results confirm that Aitken mode particles likely do not contribute to $N_d$ in polluted air masses (urban, biomass burning) at moderate updraft velocities ($w \leq 3$ m s⁻¹), but may be important in deep convective clouds. Under clean conditions, such as

in the Amazon, the Arctic, and remote ocean regions, hygroscopic Aitken mode particles can act as CCN at updrafts of $w < 1$ ms⁻¹.



## 1 Introduction

The representation of aerosol-cloud interactions in atmospheric models is challenging due to the high variability of aerosol particle loading, properties and processes on small temporal and spatial scales. Aerosol-cloud interactions include both the effects of aerosol particles on clouds by acting as cloud condensation nuclei (CCN) and the modification of aerosol due to chemical and physical cloud processing.

The interaction of aerosol particles with water vapor is described by the Köhler theory (Köhler, 1936). It combines the curvature (Kelvin) effect that describes the enhancement of the water vapor pressure above a curved particle surface and the solute (Raoult) effect that accounts for water uptake by hygroscopic particle mass which is often parameterized by the hygroscopicity parameter $\kappa$ (Petters and Kreidenweis, 2007). The maximum of the Köhler curve represents the critical supersaturation ($S_{crit}$), above which a particle of a given composition and dry size (critical diameter, $D_{crit}$) is activated and efficiently grows to a cloud droplet.

Clouds are dynamic systems where the supersaturation is continuously altered due to increasing water vapor concentration in cooling air parcels and other processes and water vapor condensation onto particles. As the supply of water vapor and the growth time scales in clouds are limited, particles and droplets may not reach their equilibrium sizes. Thus, conclusions based on equilibrium conditions as implied by Köhler theory often represent overestimates of the effect of aerosol properties on clouds, e.g., Ervens et al. (2005).

The relative importance of aerosol parameters (e.g., chemical composition ($\kappa$), dry particle diameter (D), shape of aerosol size distribution (ASD), particle number concentration ($N_a$)) and updraft velocity ($w$) on cloud properties were explored in previous sensitivity studies. Feingold (2003) showed that $N_a$ has the largest influence on the effective radius of a cloud droplet size population, followed by the geometric mean mode diameters and standard deviations of aerosol size distributions (ASDs) ($D_g$ and $\sigma_g$). A similar ranking was discussed by Pardo et al. (2019) who showed that conclusions regarding the relative importance of the aerosol properties and $w$ hold true for both the effective radius and droplet number concentration ($N_d$) with lower sensitivity of the effective radius than that of $N_d$. Other sensitivity studies also identified $w$ and $N_a$, followed by $\kappa$ and other chemical composition effects, as the most important parameters determining $N_d$ (Ervens et al., 2005; McFiggans et al., 2006; Anttila and Kerminen, 2007; Reutter et al., 2009; Ward et al., 2010) or the supersaturation in clouds (Hammer et al., 2015). Similar relative importances of $N_a$, and $w$ on the shape of the cloud droplet size distribution (CDSD) were shown (Cecchini et al., 2017).

CCN can be modified in clouds by mass addition due chemical reactions in cloud droplets and by collision-coalescence processes (e.g., Ervens, 2015). These processes are suggested to lead to a size separation of cloud-processed and interstitial particles, resulting in a gap between the Aitken and accumulation modes ('Hoppel minimum') (Hoppel et al., 1986; Cantrell et al., 1999; Feingold and Kreidenweis, 2000). It is traditionally assumed that only accumulation mode particles (D > $\sim$ 0.07 µm) undergo cloud-processing leading possibly to a broadening of this mode.

Model and observational studies challenge this assumption, as ambient conditions were identified under which supersaturation in clouds is sufficiently high to form cloud droplets on Aitken mode particles. For example, at a continental remote





background site in France, Aitken mode particles with $D \geq \sim 0.025$ µm were shown to contribute the major fraction to $N_d$ due to the absence of a significant accumulation mode (Gérémy et al., 2000). About 30% of all Aitken mode particles were observed to form cloud droplets at a background site in Finland at very low $N_a$ ($\sim$150 cm$^{-3}$) (Komppula et al., 2005). Similarly low aerosol loading was encountered above the tropical ocean, where $\leq$ 40% of Aitken mode particles were predicted to act as
CCN (Roelofs et al., 2006). Also in Arctic clouds, high contributions of Aitken mode particles to $N_d$ were predicted (Korhonen et al., 2008; Jung et al., 2018; Bulatovic et al., 2020). In marine stratocumuli off the Californian coast, CCN and droplet closure could be only achieved when contributions of Aitken mode particles to $N_d$ were taken into account in clouds with $w \geq 0.6$ m s$^{-1}$ (Schulze et al., 2020). In deep convective clouds above the Amazon ($w \leq 12$ m s$^{-1}$), it was predicted that the formation of cloud droplets on Aitken mode particles (D $\geq 0.02$ µm) might even impact the thermodynamic cloud structure by amplifying
the convective invigoration and affecting precipitation rates (Fan et al., 2018).

   Based on an intercomparison of 16 global models, it was concluded that Aitken mode particles do not significantly contribute to CCN in clouds with maximum supersaturations $S_{max}$ = 0.2% (Fanourgakis et al., 2019). Based on another global model study, Lee et al. (2013) compared the influence of 28 parameters characterizing aerosol emissions, processes and size distributions on the CCN number concentration at S = 0.3%. They identified the width of the Aitken mode as the second most
important parameter, after the dry deposition of accumulation mode particles. Chang et al. (2017) found that on a global scale, the fraction of Aitken mode particles to total CCN is negligible at S = 0.2 % while it can be significant at S = 0.4% above the continental northern hemisphere. In their later global model study, cloud supersaturation in each grid cell was calculated based on the mean vertical velocity, and $N_d$ was derived as the number of particles whose $S_{crit}$ that was approximated and compared for three cloud activation schemes (Chang et al., 2021). While the schemes mostly agreed in the $N_d$ prediction from
accumulation mode particles, large discrepancies predictions were found in the contribution of Aitken mode particles to $N_d$.

   These prior studies provide strong evidence that Aitken mode particles can cause large uncertainties in predicted aerosol-cloud interactions under conditions of low $N_a$, small fractions of accumulation mode particles to total $N_a$ and/or high $w$. In the current study, we perform simulations with an adiabatic parcel model to systematically explore the parameter ranges of aerosol properties ($N_a$, $N_{a,Ait}/N_{a,acc}$, $\kappa$, $D_{g,Ait}$, $D_{g,acc}$) and of $w$ to identify aerosol and cloud conditions, under which Aitken mode
particles contribute to $N_d$. (All parameters are defined Table A1, in the Appendix.) Our analysis results in a framework that can be used to assess under which aerosol and cloud conditions detailed information on Aitken mode particles is needed to describe their potential role in aerosol-cloud interactions.

## 2   Adiabatic parcel model

### 2.1   Model description

We use an adiabatic parcel model to examine droplet formation on a population of aerosol particles (Feingold and Heymsfield, 1992; Ervens et al., 2005). The evolution of particle and droplet sizes is described on a moving size grid. The calculation of the equilibrium saturation ($s_{eq}$) is based on Köhler theory, including the hygroscopicity parameter $\kappa$ (Petters and Kreidenweis,



2007; Rose et al., 2008).

$$s_{eq} = \left(1 + \kappa \frac{V_s}{V_w}\right)^{-1} \exp\left(\frac{4\sigma_{sol}M_w}{RT\rho_w D_{wet}}\right) \qquad \text{(E.1)}$$

whereas $D_{wet}$ is the wet particle diameter, $\sigma_{sol}$ the surface tension of the wet particle (72 mN m$^{-1}$), $\rho_s$ the density of the dry

particle, $\rho_w$ the density of pure water, $M_w$ the molecular weight of water, $R$ the constant for ideal gases and $T$ the absolute

temperature.

We note that we do not consider additional composition effects (such as surface tension suppression) as $\kappa$ represents the

effective hygroscopicity as derived from experimental data. The model includes the standard thermodynamic equations for

particle and droplet growth and the derivatives to time of temperature, saturation and pressure (Pruppacher and Klett, 2003).

These differential equations are iteratively solved within each model time step. The times steps are function of $w$ and chosen

such that they cover a vertical change of the air parcel of 0.1 m . The change in the saturation is calculated as

$$\frac{ds}{dt} = \underbrace{\Psi_1 w}_{updraft\ term}$$
$$- \underbrace{\Psi_2 2\pi \frac{\rho_w}{\rho_s} G \int D_{wet} N_a(D)(s - s_{eq}) dD_{wet}}_{condensation\ term} \qquad \text{(E.2)}$$

where $\Psi_1$ and $\Psi_2$ are functions of T and saturation ($s$). The updraft term describes the increase of $s$ due to cooling of a rising

air parcel in an adiabatic environment; the condensation term accounts for the condensation of water vapor on aerosol particles

and droplets. Particle and droplet growth is driven by the gradient between $s$ and the particle-specific $s_{eq}$:

$$\frac{dD_{wet}}{dt} \propto \frac{(s - s_{eq})}{D_{wet}} \qquad \text{(E.3)}$$

## 2.2    Model simulations

### 2.2.1    Initialization

The model is initialized below cloud at RH = 98%, T = 290 K, and p = 829 mbar. The initial ASDs consist of 545 particle size

classes in a diameter range of 0.0028 μm < D < 1.4 μm, in lognormal distributions with geometric mean mode diameters $D_{g,Ait}$

= 0.037 μm and $D_{g,acc}$ = 0.145 μm with standard deviations $\sigma_g$ = 0.5 (corresponding to 1.4 in commonly used Heisenberg fits).

Note, that different versions of lognormal fit functions are used in the literature (Pöhlker et al., 2021); a standard lognormal fit

function was applied here (Pöhlker et al., 2018).

In sensitivity tests, $D_{g,Ait}$ and $D_{g,acc}$ are shifted to 0.05, 0.06 and 0.07 μm and to 0.13 and 0.17 μm, respectively (Section

3.3.2). We define 15 model ASDs that differ (i) in the relative contributions of Aitken and accumulation mode particles ($N_{a,Ait}$,

$N_{a,acc}$) to total $N_a$ (columns I - V in Figure 1), and (ii) in the total number concentration $N_a$ (rows a - c in Figure 1). ASDs I

and V are monomodal with an accumulation or Aitken mode only; ASDs II, III and IV are bimodal with $N_{a,Ait}$ corresponding





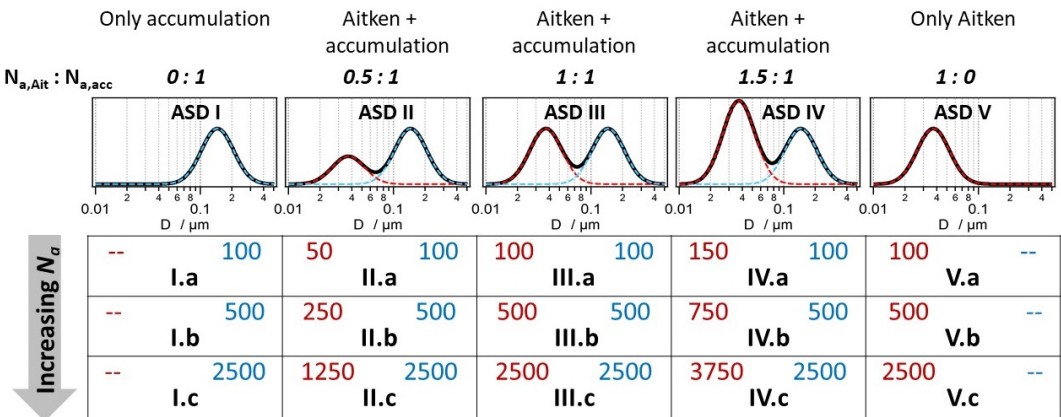

**Figure 1.** Schematic of model input aerosol size distributions (ASDs, designated I-V, a-c) with different number concentrations of Aitken mode particles ($N_{a,Ait}$, red) and accumulation mode particles ($N_{a,acc}$, blue). Particle number concentrations are given in units of cm$^{-3}$. The modal geometric mean diameters are $D_{g,Ait} = 0.037\,\mu$m and $D_{g,acc} = 0.145\,\mu$m .

to 50% (ASD II), 100% (ASD III) and 150% (ASD IV) of $N_{a,acc}$. The dashed colored lines in the upper part of Figure 1 show the overlap of the two modes near the Hoppel minimum. $N_a$ differs by a factor of five between rows for the same ASD shape.

The particle hygroscopicity is assumed to be equal in both modes; 27 values are used in the simulations to cover a range of $0.02 \leq \kappa \leq 1$. Additional tests are performed for fixed $\kappa$ values in the accumulation mode ($\kappa_{acc} = 0.1$ and $0.5$; Section 3.3.1). 30 values for the updraft velocities are applied (0.1 m s$^{-1} \leq w \leq$ 3 m s$^{-1}$), resulting in 810 simulations (27 $\kappa \times$ 30 $w$) for each

of the 15 ASDs.

### 2.2.2   Analysis of sensitivities $\xi(\kappa)$ and $\xi(N_a)$

Droplets are defined as particles with $D_{wet} \geq 3$ µm. This definition does not strictly follow Köhler theory that defines droplets as particles whose $S_{crit}$ is exceeded. The reasoning for our definition is the comparability of model results to observational studies that commonly report data from cloud probes detecting particles above a fixed size threshold, e.g., Braga et al. (2021).

As our study is intended to give guidance to future field and model studies, we use the size-based (3 µm) rather than the Köhler-based ($D_{crit}$) droplet definition. We define the smallest dry particle size on which droplet grow to $\geq 3\mu$m as $D_{min}$. The droplet number concentration $N_d$ is calculated as the cumulative particle number concentration between $D_{min}$ and the largest D (1.4 µm). We describe the predicted change in $N_d$ as a function of particle hygroscopicity ($\kappa$) as the sensitivity $\xi(\kappa)$:

$$\xi(\kappa) = \frac{\partial ln N_d}{\partial ln \kappa} \qquad\qquad\qquad (E.4)$$

For comparison to conclusions on parameter regimes as discussed in previous model sensitivity studies, we also investigate the sensitivity of $N_d$ to $N_a$

$$\xi(N_a) = \frac{\partial ln N_d}{\partial ln N_a} \qquad\qquad\qquad (E.5)$$





These definitions follow the same approach as in previous model studies that investigated the sensitivity of $N_d$ to $\kappa$ and $N_a$ for monomodal accumulation mode ASDs (e.g., McFiggans et al. (2006); Reutter et al. (2009); Ward et al. (2010); Pardo et al. (2019)).


Since $N_d$ is predicted to increase above cloud base, we perform most of our sensitivity analyses at 20 m above the height level of maximum supersaturation ($S_{max}$). In a recent $N_d$ closure study, we found not only best agreement of measured and predicted $N_d$ at this height but also of the liquid water content, independently of the pollution level of the air mass (Braga et al., 2021).

## 3  Results and Discussion


### 3.1  Vertical profiles of $N_d$, $D_{min}$ and $\xi(\kappa)$

Figure 2 shows the vertical profiles of $N_d$, $D_{min}$ and $\xi(\kappa)$ as a function of $\kappa$ (color-coding) for simulations with ASD I.b and ASD III.b at an updraft velocity of $w = 2.9$ m s$^{-1}$; complementary results for $w = 1.0$ m s$^{-1}$ and 0.2 m s$^{-1}$ are shown in Figures S1 and S2. The black lines denote the height, at which the supersaturation reaches its maximum value (Figure S3). At high $w$ and $\kappa$, $N_d$ shows significantly smaller $N_d$ for the monomodal ASD I.b than for the bimodal ASD III.b (Figure 2a and b). The $D_{min}$ values are nearly identical for the two ASDs for a given $w$ (left and right columns in Figures 2, S1 and S2) and they are inversely correlated with $N_d$. Under these conditions, $D_{min}$ reaches minimum values of ~0.05 μm, which means that all accumulation mode particles form cloud droplets ($N_d \sim N_{a,acc} = 500$ cm$^{-3}$) and $N_d$ cannot further increase for ASD I.b. In the presence of an Aitken mode, a significant fraction of highly hygroscopic Aitken mode particles grow to cloud droplets ($N_d \sim$ 620 cm$^{-3}$, Figure 2b) as $D_{min}$ is significantly smaller than the size range of the Hoppel minimum (D ~0.07 μm).



The $\xi(\kappa)$ evolution for ASD I.b (Figure 2e) repeats the trends of $D_{min}$ and mirrors those of $N_d$, i.e., $\xi(\kappa)$ is lowest for highest $\kappa$ and $w$. For ASD I.b, $\xi(\kappa)$ reaches lowest values for the highest $w$ when nearly all particles are grown to cloud droplets ($N_d \sim$ 500 cm$^{-3}$), and a decrease in $D_{min}$ does not further increase $N_d$. The difference in $\xi(\kappa)$ for ASD I.b and III.b is significant (Figure 2e, f) as $\xi(\kappa)$ for the bimodal ASD III.b is predicted to increase for $\kappa > \sim0.5$ above the level of $S_{max}$. This inversion of $\xi(\kappa)$ occurs at the height, at which Aitken mode particles start contributing to $N_d$ (Figure 2).While for the monomodal ASD I.b $\xi(\kappa)$ is predicted to continuously decrease with height, the increasing contribution of Aitken mode particles to $N_d$ leads to the opposite trend, i.e. to highest $\xi(\kappa)$ values for particles with highest $\kappa$.


Generally, the differences in the vertical profiles for ASD I.b and III.b are smaller with lower $w$ (Figures S1 and S2). At $w =$ 1 m s$^{-1}$, $N_d$ is only slightly lower for ASD I.b than for ASD III.b. The two modes overlap at the Hoppel minimum (Figure 1) and the small concentration of Aitken mode particles at this size explains the somewhat higher $N_d$ in Figure S1 a as compared to Figure S1 b. At $w = 0.2$ m s$^{-1}$, only large accumulation mode particles are activated (activated fraction, $F_{act,acc} < \sim0.5$ and $D_{min} \geq 0.12$ μm) and $\xi(\kappa)$ remains generally higher than for larger $w$. A high sensitivity implies that $D_{min}$ is in a size range, in which the ASD exhibits a steep slope where a small change in $D_{min}$ translates into a relatively large change in $N_d$.


Similar trends of $\xi(\kappa)$ with $w$ were discussed in previous studies in which it was generalized that $\xi(\kappa)$ is highest at low $w$ as only a small but significant fraction of the accumulation mode particles is activated (e.g., Moore et al. (2013); Ervens


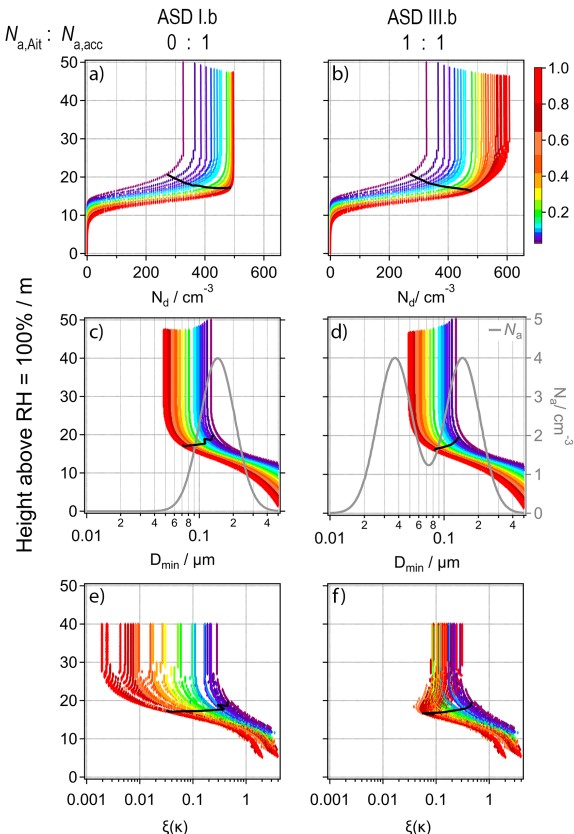

**Figure 2.** Vertical profiles of (a, b) cloud droplet number concentration ($N_d$), (c, d) dry size of smallest particles that contribute to $N_d$ ($D_{min}$), overlaid by the corresponding ASD (grey lines), and (e, f) sensitivity of $N_d$ to $\kappa$ ($\xi(\kappa)$) for an updraft velocity $w = 2.9$ m s$^{-1}$. Left and right columns show results for ASD I.b and III.b, respectively. The black lines in all panels mark the height of $S_{max}$ above the level of saturation.

et al. (2005)). Reutter et al. (2009) showed low sensitivities of $N_d$ at low $w$ for very high $N_a \sim 10{,}000$ cm$^{-3}$. In this case, the supersaturation is efficiently suppressed as the condensation term (Equation E.2) is dominated by $N_a$. As only very large particles grow to droplet sizes, $N_d$ would only include particles with D > $\sim 0.3\,\mu$m , i.e. only in a flat part of the ASD, where a small change in $D_{min}$ does not lead to a significant change in $N_d$. The increased $\xi(\kappa)$ at high $w$ for the bimodal ASD III.b
(Figure 2f) follows the same reasoning as $D_{min}$ is located at a size range where the Aitken mode exhibits approximately the same slope as at $D_{min}$ for $w = 0.2$ m s$^{-1}$. For less hygroscopic particles, $D_{min}$ is near the Hoppel minimum and thus $\xi(\kappa)$ is smaller than for high $\kappa$ (Figure 3).

For all simulations, $N_d$ is predicted to increase while $D_{min}$ decreases up to several tens of meters above the height level of $S_{max}$ (Figures 2, S1, and S2), as particles continue to grow and eventually reach the size threshold of $3\,\mu$m. The size
distributions of haze particles and droplets are shown in Figure S3 at four heights for the three $w$ and $\kappa = 0.04$, 0.3 and 0.7; the





red vertical lines indicate the size threshold (3 μm) for droplets. It can be seen that the separation of haze particles and droplets occurs at different heights, depending on $\kappa$ and $w$. Hygroscopic Aitken mode particles successively grow at high $w$ to similar droplet sizes as accumulation mode particles (orange box at the bottom of the figure), coinciding with the height at which $\xi(\kappa)$ increases. This $\xi(\kappa)$ trend is opposite to that for ASD I.b and also as found in previous studies of monomodal ASDs (Pardo
et al., 2019; Cecchini et al., 2017) that showed decreasing sensitivities to aerosol properties with height.

   If cloud droplets were defined based on $D_{crit}$, $N_d$ would be computed at the level of $S_{max}$ (black lines in Figures 1, S1, and S2) and remain constant above this height, resulting in different $N_d$ as large haze particles may not be counted if their $S_{crit}$ were not reached in cloud, but small particles whose $S_{crit}$ is exceeded might not be counted using our size-based droplet definition. A detailed comparison of predicted cloud properties applying the two droplet definitions ($D_{crit}$ versus $D_{wet} \geq 2$
μm) was performed in a previous sensitivity study (Loftus, 2018). There it was shown that predicted $N_d$ based on the two definitions show largest discrepancies at lowest $w$ and/or high $N_a$, and that the droplet size distributions are generally narrower if droplets are defined based on $D_{crit}$. Also sensitivities would be overestimated if they are computed in the unstable bottom layer of the cloud at the level of $S_{max}$. It can be concluded sensitivity studies, in which a droplet definition based on $D_g$ is applied will not only lead to higher absolute values of $\xi(\kappa)$ but at the same time, also to an underestimate of the sensitivity to
$\kappa$ as the predicted differences of $\xi(\kappa)$ for different $\kappa$ values are much smaller (e.g., Reutter et al. (2009)). Such studies might thus lead to biased conclusions if they are applied to ambient $N_d$ measurements that are based on fixed size thresholds to discriminate cloud droplets.

### 3.2   Sensitivity to aerosol properties: $\xi(\kappa)$ and $\xi(N_a)$

#### 3.2.1   Dependence of $\xi(\kappa)$ on $D_{min}$ (ASD III.b)

In the following, we investigate more generally the parameter ranges, at which Aitken mode particles affect sensitivities and $N_d$ in clouds. Our discussion will be limited to cloud conditions at a height of 20 m above $S_{max}$, i.e., when $N_d$ has reached a constant value. Figure 3 depicts the $\xi(\kappa)$ values resulting from the 810 simulations for ASD III.b as a function of $\kappa$ and $w$. The contour lines are color-coded by $0 \leq \xi(\kappa) \leq 1$. Parallel to the axes, six lines are marked for three $\kappa$ values (vertical lines at $\kappa$ = 0.7 (1), 0.3 (2), and 0.04 (3)), and three updraft velocities (horizontal lines at $w$ = 2.9 m s$^{-1}$ (4), 1.0 m s$^{-1}$ (5) and 0.2 m s$^{-1}$ (6)).
As $N_d$, $D_{min}$ and $\xi(\kappa)$ are closely related (Section 3.1), we can ascribe every $\xi(\kappa)$ value in Figure 3a to a $D_{min}$ value for each combination of $w$ and $\kappa$. In Figure 3b, the three blue lines show $\xi(\kappa)$ as a function of $D_{min}$ (right axis), for a single $\kappa$ (0.04, 0.3 or 0.7) and for the full range of $w$. The left ends of the lines indicate $D_{min}$ for the simulation $w$ = 3 m s$^{-1}$, the right ends of the same lines correspond to $D_{min}$ for the same $\kappa$ and $w$ = 0.1 m s$^{-1}$. The ranges of $D_{min}$ are also indicated as the color-matched axes between the figure panels b and c. In the same way, Figure 3c shows $D_{min}$ for $\xi(\kappa)$ resulting from
simulations for three single values of $w$ and the full range of $\kappa$ with the $D_{min}$ ranges indicated by the red, orange and yellow lines above the figure.

   In line with the results in Figures 2, $\xi(\kappa)$ and $D_{min}$ are highest for small $\kappa$ and $w$ and decrease with increasing $w$ (Figure 3b). Analogous trends are shown in Figure 3b for the $\xi(\kappa)$ values as a function of $D_{min}$ along the horizontal lines in Figure



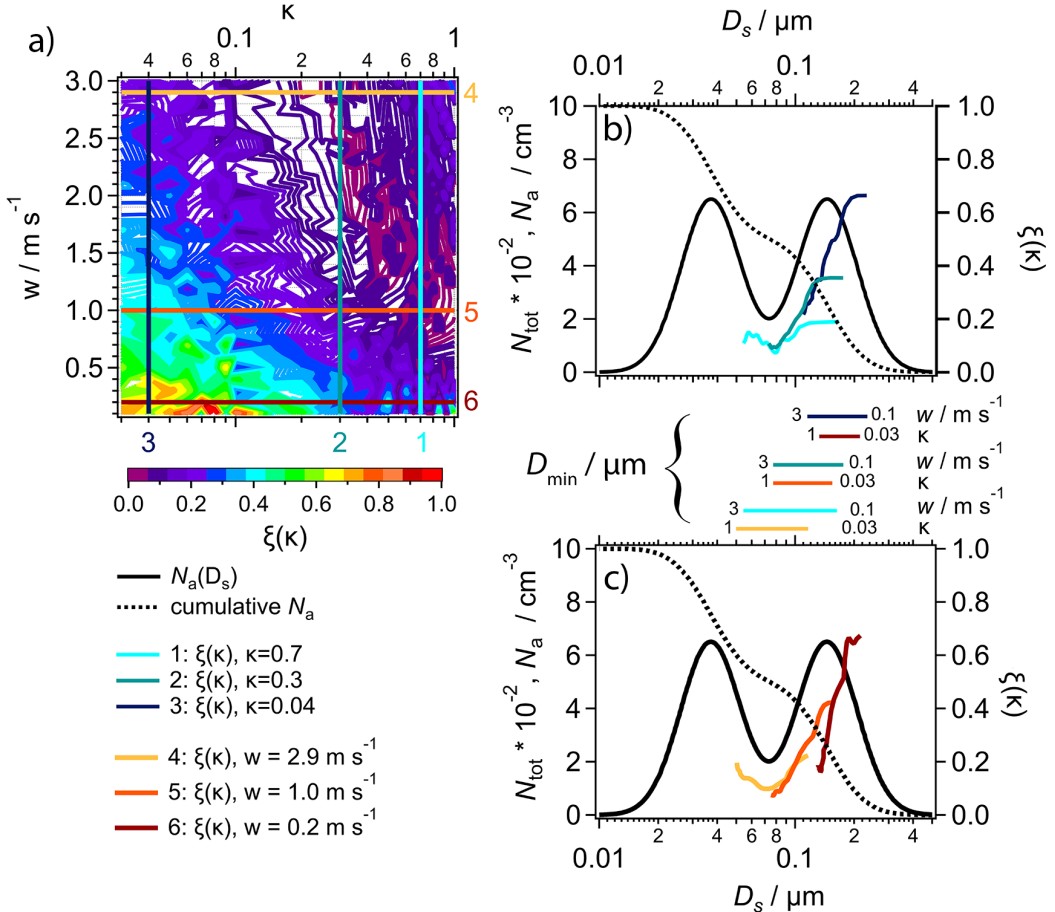

**Figure 3.** a) Sensitivity of cloud droplet number concentration to aerosol hygroscopicity ($\xi(\kappa)$) as a function of $\kappa$ and $w$ for ASD III.b at 20 m above $S_{max}$. b, c): $\xi(\kappa)$ (right axis) as a function of $D_{min}$ for b) 1. $\kappa = 0.7$, 2. $\kappa = 0.3$, 3. $\kappa = 0.04$ (vertical lines in a)), and c) 4. $w = 2.9$ m s$^{-1}$, 5. $w = 1.0$ m s$^{-1}$, 6. $w = 0.2$ m s$^{-1}$ (horizontal lines in a)). The $D_{min}$ range as a function of $w$ or $\kappa$, respectively, is indicated by the colored axes between b) and c). ASDs and cumulative ASDs are shown as black solid and dashed lines (left axis).

3a. The $\xi(\kappa)$ lines for the three $\kappa$ or three $w$ values, respectively, overlap as $\xi(\kappa)$ depends on the slope of the ASD at $D_{min}$.
Different combinations of $\kappa$ and $w$ can result in identical $D_{min}$ values and thus yield the same $\xi(\kappa)$ values (contour lines of identical color in Figure 3a). When $D_{min}$ is near the Hoppel minimum, a change in $D_{min}$ by either of the parameters does not lead to a significant change in $N_d$ leading to low $\xi(\kappa)$ values. Accordingly, $\xi(\kappa)$ increases when $D_{min}$ reaches sizes smaller than the Hoppel minimum. Overall, $\xi(\kappa)$ as a function of $D_{min}$ traces the shape of the ASD (black line in Figure 3b and c). The combinations of $w$ and $\kappa$ for which an increase in $\xi(\kappa)$ with a decrease in $D_{min}$ is predicted are those in the upper right
corner of Figure 3a.



The $D_{min}$ ranges, covered by a variation of $w$ or $\kappa$ by the same factor ($\sim 30$) (colored lines between 3b and c), are similar. This suggests that an equal change in either of the parameters affects $D_{min}$ to the same extent. However, this relationship cannot be generalized. Figures S5 and S6 show equivalent results to those in Figure 3, but for $N_a$ = 200 cm$^{-3}$ and $N_a$ = 5000 cm$^{-3}$, respectively. Smaller $N_a$ implies a smaller condensation term (Equation E.2), resulting in a higher supersaturation, which allows also smaller particles to grow to cloud droplets. Thus, for $N_a$ = 200 cm$^{-3}$, the $D_{min}$ ranges are shifted to smaller values, resulting in higher $N_d$ as a larger fraction of the (cumulative) ASD is activated (Figure S5 b and c). Accordingly, the $D_{min}$ ranges move to larger values for $N_a$ = 5000 cm$^{-3}$ (Figure S6 b and c).

The comparison of Figures 3, S5 and S6 reveals that not only the ranges of $D_{min}$ values are shifted as a function of $N_a$, but also that their widths differ depending on $N_a$. The $D_{min}$ ranges are widest for $N_a$ = 200 cm$^{-3}$ which implies that for these conditions $N_d$ is most sensitive to $\kappa$ and $w$, as a change in these parameters causes a significant change in $D_{min}$ and $N_d$. Correspondingly, for $N_a$ = 5000 cm$^{-3}$, a change in $w$ or $\kappa$ only leads to a small change in $D_{min}$. While such a shift in $D_{min}$ only leads to small change in the activated fraction, it translates into a relatively large difference in $N_d$, resulting in high $\xi(\kappa)$. This analysis demonstrates that the similarity in the $D_{min}$ ranges in Figure 3b and c resulting from a change in $\kappa$ or $w$ by the same factor are coincidental and should not be generalized to all conditions as the relative sensitivities to $\kappa$ and $w$ depend on $N_a$. However, it also shows that conditions exist under which $N_d$ is similarly sensitive to $w$ and $\kappa$ and both parameters need to be taken into account to accurately predict $N_d$.

### 3.2.2 Sensitivity regimes of $\xi(N_a)$ for mono- and bimodal ASDs

Previously, sensitivity of $N_d$ to $N_a$ and to $w$ were presented in terms of aerosol ($N_a$)- and updraft ($w$)-limited regimes (Reutter et al., 2009, 2014; Chang et al., 2015). The $N_a$-limited regime is characterized by high activated fractions, i.e., when an increase in $N_d$ can be only caused by an increase in $N_a$ and $N_d$ depends linearly only on $N_a$; the $w$-limited regime occurs for small activated fractions where an increase in $w$ leads to sufficient decrease in $D_{min}$ to increase $N_d$. Our analysis in Section 3.2.1 suggests that for wide parameter ranges, $N_d$ is similarly sensitive to $\kappa$ and $w$. To discuss these results in the context of $N_a$- and $w$-limited regimes, we explore the sensitivity of $N_d$ to $N_a$, $\xi(N_a)$, as a function of $w$ for $\kappa$ = 0.7 for ASD I, III and V with 500 cm$^{-3} \leq N_a \leq$ 5000 cm$^{-3}$.

Figure 4a shows $N_d$ for ASD I as a function of $N_a$ and $w$ and confirms the thresholds between the regimes as suggested by Reutter et al. (2009), i.e., $N_a$-limitation above $w/N_a > \sim10^{-3}$ m s$^{-1}$ cm$^3$ and $w$-limitation below $w/N_a < \sim10^{-4}$ m s$^{-1}$ cm$^3$, with a transitional regime in-between. Accordingly, $\xi(N_a)$ approaches unity in the $N_a$-limited regime when nearly all particles are activated (upper left corner of Figure 4b) and even exceeds this value at high $N_a$ and low $w$ (bottom right corner).

The presence of an Aitken mode, in addition to an accumulation mode (ASD III; Figure 4c), $N_d$ is not limited by $N_a$ under the same $N_a$ and $w$ conditions as for ASD I since only a small fraction of the Aitken mode is activated. Thus, the transitional regime is extended to a broader parameter space as $\xi(N_a)$ does not show a constant value of unity as it should in an $N_a$-limited regime. As at high $N_a$ and low $w$, only accumulation mode particles contribute to $N_d$, the contours for the $w$-limited regime do not differ between Figures 4b and d.

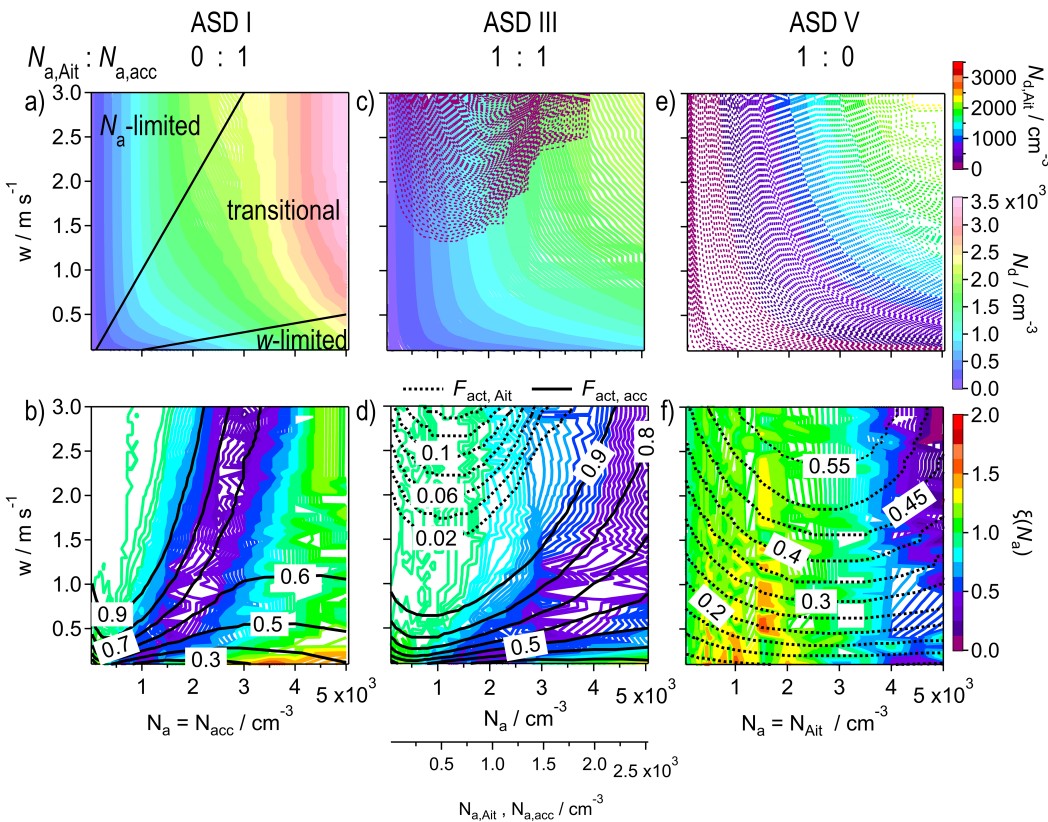

**Figure 4.** Upper panels: $N_d$ as a function of $w$ and $N_a$ for ASD a) I, c) III and e) V. Lighter colors: $N_{d,acc}$, darker colors: $N_{d,Ait}$. Bottom panels: Corresponding $\xi(N_a)$ for ASD b) I, d) III and f) V. Solid black lines show contours of activated fraction of the accumulation mode ($F_{act,acc}$), dashed lines show contours of activated fraction of the Aitken mode ($F_{act,Ait}$).

For the monomodal ASD V (Aitken mode), the activated fraction reaches at most ∼0.6 (Figure 4e, f); thus, the aerosol-

limited regime is not reached. However, for $N_a >$∼1000 cm⁻³ and $w <$ ∼1.0 m s⁻¹, $N_d$ is linearly dependent on $w$ and in-
dependent of $N_a$, which implies a $w$-limited regime for $w/N_a <$ ∼10⁻³ m s⁻¹ cm³, i.e. shifted by an order of magnitude as
compared to ASD I. Accordingly, $\xi(N_a) \geq 1$, similar to the $\xi(N_a)$ values reached for much higher $N_{a,acc}$ and lower $w$ for the
monomodal accumulation mode ASD I. In summary, based on these trends of $\xi(N_a)$ and those of $\xi(\kappa)$ in Section 3.2.1, the
following conclusions can be drawn:

255        – The $N_a$ and $w$-limited regimes are dependent on particle size and thus the $w/N_a$ limits are different for monomodal
             Aitken vs accumulation mode ASDs and also for monomodal vs bimodal ASDs.

           – The sensitivities of $N_d$ to ASD parameters ($N_a$, $D_g$, $\kappa$) and to $w$ depend on their value combinations.





– Under most $w$ and $N_a$ conditions as considered here for bimodal ASDs, the $N_a$-limited regime is not reached, as they cover the transitional regime.

– The equivalency of a change in $w$ and $\kappa$ to affect $D_{min}$ (Figure 3) implies that a $\kappa$-sensitive regime could be equally defined and taken into account as a $w$-sensitive regime when exploring sensitivities of $N_d$.

### 3.2.3   Sensitivities $\xi(\kappa)$ for different aerosol size distributions

The parameter ranges of $w$, $\kappa$ and $N_a$ considered in our simulations in the presence of an Aitken mode constrain transitional or $w$-limited regimes, in which $N_d$ can be equally influenced by $w$ and $\kappa$. Therefore, we explore in detail the $w$,$\kappa$ combinations,
above which Aitken mode particles significantly affect $N_d$ and $\xi(\kappa)$. Figure 5 shows $\xi(\kappa)$ contour plots for all cases as defined in Figure 1; the middle column (ASD III) repeats Figures 3a, S5 a and S6 a. As the supersaturation and activated fractions are closely related to $\xi(\kappa)$, the corresponding figures for the 15 cases are shown in Figures S7 and S8.

The $\xi(\kappa)$ values for the monomodal accumulation mode ASD I are shown in the first column of Figure 5 (a: $N_a$ = 100 cm$^{-3}$, b: 500 cm$^{-3}$ and c: 2500 cm$^{-3}$). The white regions in the upper right part of the figures mark the parameter spaces above
which $\xi(\kappa) \sim 0$, i.e., the $N_a$-limited regime. At high $N_a$, this space is shifted to higher $\kappa$ and $w$, in agreement with the trends of larger $D_{min}$ at higher $N_a$ (Section 3.1.1). The trends in $\xi(\kappa)$ along Columns II, III and IV show the effect on $\xi(\kappa)$ due to increasing $N_{a,Ait}$ / $N_{a,acc}$ (0.5, 1.0, 1.5) for a fixed $N_{a,acc}$ in each row. The $\xi(\kappa)$ contours in the parameter space, at which only accumulation mode particles contribute to $N_d$ do not significantly change for a given $N_{a,acc}$ (bottom left corners of panels ASD I - IV). However, with increasing $N_{a,Ait}/N_{a,acc}$, the absolute $\xi(\kappa)$ values increase, i.e., $\sim 0.15 \le \xi(\kappa) \lesssim 0.25$ for
the major part of panels IV.a and IV.b whereas $0 \le \xi(\kappa) \lesssim 0.25$ in panels II.a and II.b. A higher $N_{a,Ait}/N_{a,acc}$ causes the range to narrow, at which $\xi(\kappa)$ shows a minimum, while the $w$,$\kappa$ combinations for minimum $\xi(\kappa)$ are not significantly shifted (ASD I - IV within a each row in Figure 5). This trend in $\xi(\kappa)$ is caused by the higher $N_{a,Ait}$ near the Hoppel minimum with increasing $N_{a,Ait}/N_{a,acc}$, and thus higher $N_d$ when $D_{min} \sim 0.07$ μm.

At the highest $N_a$, the $\xi(\kappa)$ patterns do not show any significant difference (I.c to IV.c in Figure 5). As under these conditions
for $\kappa$, $w$ and $N_a$ only accumulation mode particles are activated, the presence of the Aitken mode does not affect $\xi(\kappa)$ within the $\kappa$ and $w$ ranges considered here. If our scales were extended to updraft velocities of several meters per second as encountered in deep convective clouds, Aitken mode particle particles may activate even if $N_{a,acc} \ge 2500$ cm$^{-3}$ resulting in similar contour patterns as for lower $N_a$ and the $w$ ranges considered here. However, in polluted air masses eventually a saturation effect in terms of droplet formation is reached above which $N_d$ and effective radii do not significantly change as it was shown for
convective clouds in the Amazon region (Polonik et al., 2020).

At first sight, the $\xi(\kappa)$ patterns for ASD V are very different to those of the other ASDs (columns I-IV vs V in 5). These apparently different trends can be reconciled based on the discussion in Sections 3.2.1. and 3.2.2: The higher $\xi(\kappa)$ for highly hygroscopic particles (red area at the bottom right corner in Figure 5V.a-c) implies that at low $w$, a large number of particles with high $\kappa$ form droplets whereas $N_d$ is smaller for less hygroscopic particles. This suggests that $D_{min}$ for high $\kappa$ is located





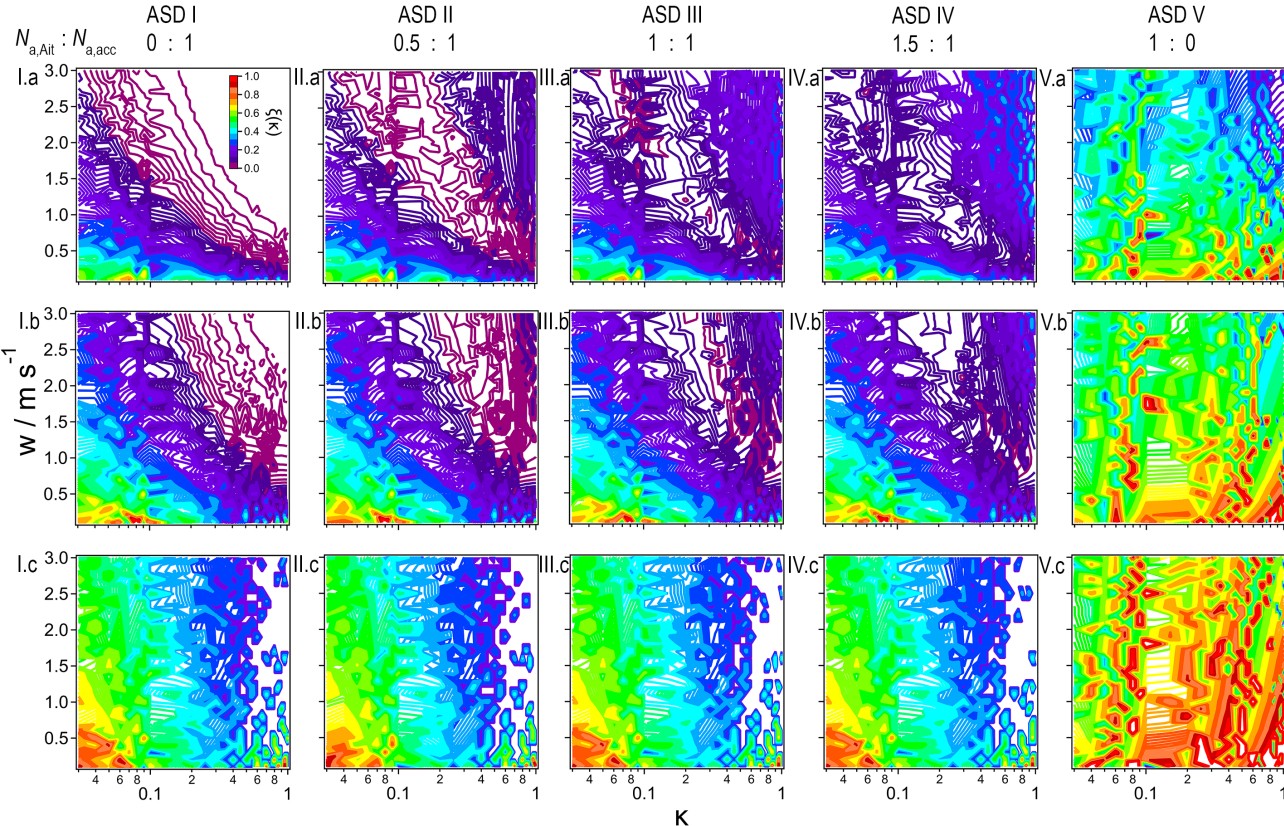

**Figure 5.** Sensitivity of $N_d$ to $\kappa$ ($\xi(\kappa)$) as a function of $w$ and $\kappa$ for ASDs I.a - V.c (Figure 1). Contour plots ($0 \leq \xi(\kappa) \leq 1$, color-code in panel a) are based on 810 model simulations assuming 30 different values of $w$ and 27 different values of $\kappa$ for each ASD. Column ASD III repeats panels a) in Figures 3, S5 and S6.

in a 'steep' part of the ASD with relatively high $N_{a,Ait}$, and a decrease in $\kappa$ increases $D_{min}$ such that it is in the flat part of the ASD ($D \sim 0.1\,\mu m$).

For ASD I, such situations are not encountered for the chosen parameter ranges. Even for the lowest $\kappa$ and $w$, a significant fraction of $N_{a,acc}$ is activated, and any change in $D_{min}$ due to a change in $w$ or $\kappa$ shifts it along the 'steep part' of the cumulative ASD. If we performed simulations for higher $N_{a,acc}$, i.e., for conditions typical for a $w$-limited regime, $D_{min}$ would be shifted to even larger sizes than shown in Figure S6 b and c, resulting in very small $N_d$ and $\xi(\kappa)$.

As discussed in Section 3.2.2., the conditions for ASD V can be described as a $w$- and/or $\kappa$-limited regime that it is characterized by a relatively low $w/Na$ (or $\kappa/N_a$) ratio. Using ASD I, similar $\xi(\kappa)$ patterns as in Figure 5V.a could be obtained using ASD I.a for smaller $\kappa$ and/or $w$ or higher $N_a$ than considered here. This can be seen in the upper row of Figure 5 where panel V.a is apparently a continuation of panel I.a to smaller $\kappa$ values. However, the match of the corresponding panels (V.b to I.b or V.c to I.c, respectively) is not as perfect which suggests different sensitivities of $\xi(\kappa)$ to $D_g$ and $\kappa$, depending on $N_a$. Thus, not





only are the limits of the $w$-limited regime depending on $D_g$ (i.e., accumulation or Aitken mode) but also the parameter range over which the regimes occur.

The comparison of the influence on $\xi(\kappa)$ due to increasing $N_{a,Ait}$ (V.a, b, c in Figure 5) and on $\xi(N_a)$ (Figure 4f) shows that $\xi(\kappa)$ is increasing whereas $\xi(N_a)$ is decreasing. Therefore, when high concentrations of Aitken mode particles dominate the

ASD, $N_d$ is highly sensitive to $\kappa$ and $w$, and less to $N_{a,Ait}$. While in the presence of accumulation mode dominated ASDs such $w$- and $\kappa$-sensitive conditions might be only encountered in highly polluted air masses (e.g., biomass burning), they can occur for Aitken mode dominated ASDs at much lower aerosol concentrations. This shift in sensitivities might partially explain the large uncertainties in cloud properties predicted in global model studies when Aitken mode particles significantly influence $N_d$ and other cloud properties (e.g., Lee et al. (2013); Chang et al. (2021)).

### 3.3    Dependence of $\xi(\kappa)$ on $\kappa_{acc}$ and $D_g$

#### 3.3.1    Influence of constant $\kappa_{acc}$ on $\xi(\kappa)$

While we have assumed so far that both Aitken and accumulation modes have the same $\kappa$, such conditions are rarely encountered in the atmosphere. They might occur, for example, when sea-salt contributes significantly to both modes (Wex et al., 2016). However, more frequently the accumulation mode consists of material of higher hygroscopicity as it accumulates sulfate and other compounds during cloud processing and other ageing processes. Continental accumulation modes typically exhibit

values in a range of $0.1 < \kappa_{acc} \lesssim 0.5$ with an average value of $\kappa_{acc} \sim 0.3$ (Andreae and Rosenfeld, 2008). The Aitken mode is comprised of fresher, less hygroscopic particles with $0 < \kappa \lesssim 0.05$ in urban and continental air masses, corresponding to hygroscopic growth factors $\leq 1.1$ at RH = 90 %. Aged Aitken mode particles are more hygroscopic, with $\kappa \sim 0.3$ in the free troposphere and $\kappa \sim 0.6$ in remote marine air (McFiggans et al., 2006).Similar trends in the hygroscopicity of the two modes

were also observed during the wet season in the Amazon with $\kappa_{Ait} \sim 0.1$ and $\kappa_{acc} \sim 0.2$ (Zhou et al., 2002; Gunthe et al., 2009). In a global model study, fairly large differences were predicted above oceans ($\kappa_{Ait} \sim 0.5$, $\kappa_{acc} \sim 1$), and more similar values for both modes above continents ($\sim 0.3 \leq \kappa \lesssim 0.8$; Chang et al. (2017)).

To explore situations with $\kappa_{acc} \neq \kappa_{Ait}$, we repeat the simulations for ASD III.b but use a single value of $\kappa_{acc}$, whereas the full range of $0.02 \leq \kappa_{Ait} \leq 1$ is applied (Figure 6a). For low $w$ and $\kappa$, $\xi(\kappa_{Ait})$ is zero (white space in Figure 6c and d)

because in this parameter range droplets only form on accumulation mode particles and a change in $\kappa_{Ait}$ does not affect $N_d$. When $\kappa_{Ait} > \kappa_{acc}$, small accumulation mode particles may not become activated whereas more hygroscopic (but smaller) Aitken mode particles sufficiently grow and contribute to $N_d$. Thus, the total $N_d$ is the sum based on two separate $D_{min}$ values for Aitken and accumulation modes, respectively. The activated fractions for the two simulations are compared in Figure S8; $F_{act,acc}$ appears as horizontal lines as it is independent of $\kappa_{Ait}$; only at high $\kappa_{Ait}$ values, there are small deviations from this

behavior as very hygroscopic Aitken mode particles may sufficiently suppress the supersaturation and prevent efficient growth of accumulation mode particles. The large overlap of the activated fractions from both modes in Figure S8 a ($\kappa_{acc} = 0.1$) demonstrates that at $w \gtrsim 1$ ms$^{-1}$, Aitken mode particles with $\kappa_{Ait} \gtrsim 0.3$ may grow to droplet sizes even though only $\sim 70\%$ of accumulation mode particles are activated.





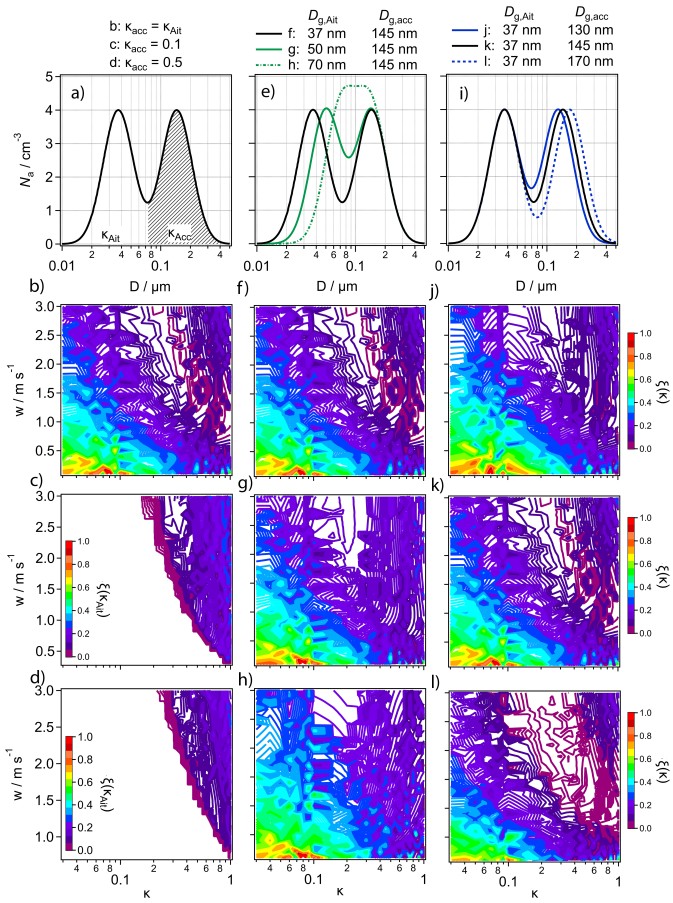

**Figure 6.** Sensitivity of $N_d$ to $\kappa$ ($\xi(\kappa)$) as a function of $w$ and $\kappa$ for variable ASDs and aerosol properties as outlined in the top panels. a) Schematic ASD with constant $\kappa_{acc}$; b) identical to Figure 5 III.b, c) $\xi(\kappa_{Ait}$, assuming $\kappa_{acc} = 0.1$, d) $\xi(\kappa_{Ait}$, assuming $\kappa_{acc} = 0.5$, e) ASDs with different $D_{g,Ait}$ in addition to ASD III, f) identical to Figure 5 III.b, g) $\xi(\kappa)$ for $D_{g,Ait} = 0.05$ μm, h) $\xi(\kappa)$ for $D_{g,Ait} = 0.07$ μm, i) ASDs with different $D_{g,acc}$ in addition to ASD III, j) $\xi(\kappa)$ for $D_{g,acc} = 0.13$ μm, k) identical to Figure 5 III.b, l) $\xi(\kappa)$ for $D_{g,acc} = 0.17$ μm

The similar $\xi(\kappa_{(Ait)})$ values in Figure 6b-d show their weak dependence on $\kappa_{acc}$. The supersaturation is largely controlled

by $N_{a,acc}$ resulting in very similar values over the full $\kappa$ range, independent of the presence of an Aitken mode (Figure S7 I.b - IV.b). When $\kappa_{acc} = 0.1$, droplet formation on Aitken mode particles occurs for slightly lower values of $\kappa$ and $w$ as compared to the case with $\kappa_{acc} = 0.5$ (Figure 6c and d). More hygroscopic accumulation mode particles efficiently suppress the supersaturation and prevent smaller (Aitken mode) particles from efficient growth. Thus, for the same $w,\kappa$ combinations, $F_{act,Ait}$ is smaller when $\kappa_{acc} = 0.5$ compared to $\kappa_{acc} = 0.1$ (Figure S8).

Similar feedbacks of the two modes on cloud properties were described in previous model sensitivity studies that showed more Aitken mode particles to activate in the presence of less hygroscopic accumulation mode particles (Kulmala et al., 1996).





In the latter study, soluble and insoluble mass fractions were used as proxies of particle composition, and it was shown that CCN activation can be parameterized by the soluble mass, which may be higher in large, soluble (hygroscopic) Aitken than in small, less soluble accumulation mode particles. A significant contribution of Aitken mode particles to $N_d$ (> 50%) was observed at
a background site in northern Finland with average activated fractions of $F_{act,acc}$ ~87% and $F_{act,Ait}$ ~30% (Komppula et al., 2005). While such observations could be equally explained by externally mixed aerosol, this would result in the same effects during cloud processing: Efficient formation of mass (e.g. sulfate) in droplets formed on Aitken mode particles that lead to a narrowing of the Hoppel minimum rather than to a widening. In a global model study, it was demonstrated that efficient sulfate formation in such droplets could contribute several percent ($\gtrsim$5%) to the global sulfate budget (Roelofs et al., 2006).

### 3.3.2  Influence of $D_{g,Ait}$ and $D_{g,acc}$ on $\xi(\kappa)$

The parameters commonly used to characterize lognormal ASDs, $D_g$ and $\sigma_g$, were identified as the most important aerosol parameters in affecting cloud properties (e.g., Feingold (2003); Ervens et al. (2005); Reutter et al. (2009); Ward et al. (2010); Anttila et al. (2012)). To compare their importance for accumulation and Aitken modes, we vary $D_{g,Ait}$ and $D_{g,acc}$ within the range of observed values. Our base case values ($D_{g,Ait}$ = 0.037 µm, $D_{g,acc}$ = 0.145 µm, Figure 1) are typical for oceanic
aerosol (Wex et al., 2016). In addition, we apply $D_{g,acc}$ = 0.13 µm and 0.17 µm for continental aerosol (Pöhlker et al., 2016, 2018). The size of Aitken mode particles strongly depends on their ageing state; it is larger for continental aerosol than above the ocean (e.g., Birmili et al. (2001); Heintzenberg et al. (2004); Pöhlker et al. (2016, 2018)). In sensitivity tests, we assume $D_{g,Ait}$ = 0.05 µm and 0.07 µm while keeping $D_{g,acc}$ = 0.145 µm.

We use ASD III.b as the reference case; it is shown together with the $D_g$-shifted ASDs in Figure 6e and i. The panels below
the ASDs show the effect on $\xi(\kappa)$ of increasing $D_{g,Ait}$ (f, g, h) and $D_{g,acc}$ (j, k, l). With increasing $D_{g,Ait}$, the parameter space at which $\xi(\kappa)$ exhibits minimum values is shifted to lower $w$ and $\kappa$ values. The lowest values of $\xi(\kappa)$ (~0) are predicted for the smallest $D_{g,Ait}$ whereas $\xi(\kappa)$ ~0.2 for most of the $w,\kappa$ space above which Aitken mode particles are activated. This is in agreement with our interpretation of Figure 3 that the dependence of $\xi(\kappa)$ on $D_{min}$ traces the ASD shape. With $D_{g,Ait}$ = 0.07 µm, there is no $w,\kappa$ space, in which $\xi(\kappa)$ shows a distinct minimum as both modes largely overlap (green dotted line
in Figure 6e). As $D_{min}$ is determined by the supersaturation, which, in turn, is largely controlled by the accumulation mode properties ($N_{a,acc}$, $D_{g,acc}$, $\kappa$), a shift of $D_{g,Ait}$ to larger sizes moves $D_{min}$ to a different part of the ASD. For example, while $D_{min}$ ~0.06 µm only leads to a small activated fraction of Aitken mode particles ($F_{act,Ait}$ < 0.1) if $D_{g,Ait}$ = 0.037 µm, it would be > 0.5 with $D_{g,Ait}$ = 0.07 µm.

Accordingly, the trends in $\xi(\kappa)$ for a change in $D_{g,acc}$ (Figure 6j, k, l) can be explained: The Hoppel minimum is widest for
the ASD with $D_{g,acc}$ = 0.17 µm which is reflected by the large space, in which $\xi(\kappa) \sim 0$ (Figure 6l). However, unlike the shift of the range in which $\xi(\kappa)$ shows a minimum to lower $w,\kappa$ values for increasing $D_{g,Ait}$, the $w,\kappa$ space becomes only broader for larger $D_{g,acc}$ but barely changes its position. It can be concluded that $D_{g,acc}$ is of minor importance as compared to $D_{g,Ait}$ for the $w,\kappa$ parameter space above which Aitken mode particles contribute to $N_d$.





## 4 Updraft and hygroscopicity regimes of Aitken mode CCN activation

Our sensitivity studies have shown that for bimodal (Aitken and accumulation mode) ASDs, the $w,\kappa$ combinations resulting in
$\xi(\kappa)$ minimum values can be used as a criterion of conditions under which Aitken mode particles contribute to $N_d$. An increase
in $w$ or $\kappa$ decreases $D_{min}$ to sizes smaller than the region of the Hoppel minimum.

To provide a general framework of our model results, we extract from the sensitivity simulations the $w,\kappa$ combinations as
thresholds, at which $F_{act,Ait}$ = 0.05 as a threshold, above which Aitken mode particles may significantly contribute to the
total droplet number concentration. Figure 7a and c presents a selection of our model ASDs, together with additional ASDs
($N_{a,Ait}/N_{a,acc} \sim 10$ and $\sim 0.1$) to further map out the parameter space. In the bottom panels, the $w,\kappa$ lines are summarized
from each simulation that correspond to $F_{act,Ait} = 0.05$ (Figure 7b, d).

The black lines in Figure 7 show two ASDs with equal contributions of Aitken and accumulation mode to $N_a$ (1000 cm⁻³,
ASD III.b). For ASDs of these $D_g$ and $N_a$, any combination along the lines yields $F_{act,Ait} \geq 0.05$, such as $w \geq 1.5$ m s⁻¹ and
$\kappa_{Ait} \sim 1$, or $\kappa \geq 0.3$ and $w \sim 3$ m s⁻¹, respectively. The red lines in Figure 7b mark the shift of the $w,\kappa$ as a function of $N_a$
and $N_{a,acc}/N_{a,Ait}$: Decreasing $N_a$ of both modes by a factor 5 moves the line to much lower $w$ and $\kappa$ values, such that Aitken
mode particles with $\kappa \sim 0.4$ may be activated at $w \gtrsim 1$ m s⁻¹ and more hygroscopic Aitken mode particles at even lower $w$
(red dotted line). When the ASD is dominated by an Aitken mode ($N_{a,Ait} \sim 10 N_{a,acc}$), Aitken mode particles with $\kappa_{Ait} \geq$
0.1 will be activated at $w \geq 1$ m s⁻¹ (red dashed line). Conversely, when $N_{a,acc} \sim 10 N_{a,Ait}$ with $N_{a,Ait} = 55$ cm⁻³, the $w$ and
$\kappa$ values are nearly outside the ranges of our $\kappa$ and $w$ scales (red solid line). Results for even higher $N_a$ (2500 - 6750 cm⁻³)
are consequently not included in the figure because under these conditions, the efficient suppression of the supersaturation
by the high $N_{a,acc}$ preventing Aitken mode particles from being activated within the considered parameter ranges of $w$ and $\kappa$.
Obviously, for wider $w$ ranges as relevant for pyrocumuli or other highly convective cloud systems, corresponding thresholds
and $w,\kappa$ combinations could be derived on extended axes.

The purple lines in 7c correspond to results for constant $N_{a,acc} = 500$ cm⁻³ and for $N_{a,Ait}$ being 0.5 and 1.5 times that of
the accumulation mode (ASD II.b, IV.b). The fact that they do not show any noticeable difference to results using ASD III.b
demonstrates that the ratio $N_{a,Ait}/N_{a,acc}$ - if $N_{a,acc}$ is approximately constant - does not significantly impact the position of
the $w,\kappa$ line.

In Figure 7d, we show the effects of $D_{g,Ait}$ and $D_{g,acc}$ on the $w,\kappa$ space. A change in $D_{g,acc}$ from 0.145 μm to 0.13 μm or
to 0.17 μm has a negligible effect (blue solid and dotted lines). The low sensitivity of the $w,\kappa$ line position to $D_{g,acc}$ suggests
that it is applicable to continental and marine air masses, largely independent of $D_{g,acc}$. However, an increase in $D_{g,Ait}$ from
0.037 μm (ASD III.b) to 0.05 μm, 0.06 μm and 0.07 μm (green lines) significantly reduces the $w$ and $\kappa$ values that are required
to activate Aitken mode particles. Thus, ageing processes might efficiently increase $D_{g,Ait}$ to those of CCN.

Figure 7b and d represents a simple scheme that can be applied to estimate whether Aitken mode particles contribute to $N_d$
under ambient conditions in various air masses and cloud types. While we did not explore all relevant combinations of the
parameters that are commonly used to characterize ASDs ($N_a$, $N_{a,Ait}/N_{a,acc}$, $D_{g,acc}$, $D_{g,Ait}$), the ranking of their relative
importance in determining the position of the $w,\kappa$ line is expected to hold generally true.





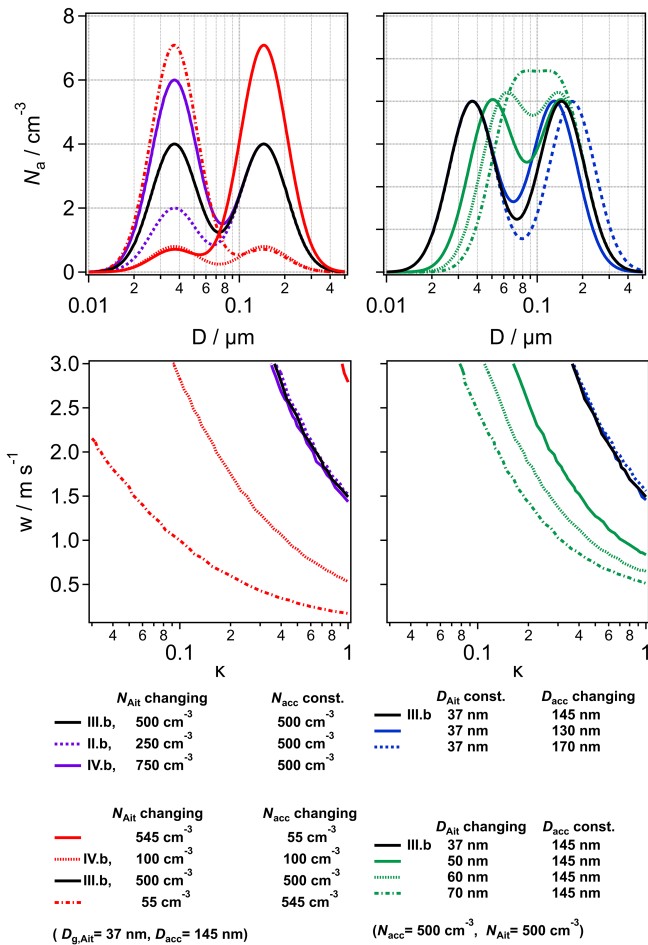

**Figure 7.** Updraft and hygroscopicity regimes where Aitken mode particles are relevant as CCN. a) Aerosol size distributions with different $N_a$ and $N_{a,Ait}$ / $N_{a,acc}$, b) Lines indicate the $w$, $\kappa$ combinations for which $F_{act,Ait}$ = 0.05 using the corresponding ASDs in a) (same color), c) Aerosol size distributions with different $D_{g,Ait}$ and $D_{g,acc}$, d) Lines indicate the $w$, $\kappa$ combinations for which $F_{act,Ait}$ = 0.05 using the corresponding ASDs in c) (same color).

## 5   Summary and conclusions

Previous field and model studies suggested that not only accumulation, but also - under specific conditions - Aitken mode
particles increase cloud droplet number concentrations and be involved in aerosol-cloud interactions. However, the conditions, under which the Aitken mode significantly contributes to cloud droplet number concentration ($N_d$) had not been fully constrained.

Using an adiabatic parcel model, we systematically investigated the conditions under which Aitken mode particles contribute to $N_d$ for wide ranges of aerosol size distribution (ASD) parameters (particle number concentrations of accumulation and





Aitken modes, $N_{a,acc}$, $N_{a,Ait}$, mode diameters, $D_{g,acc}$, $D_{g,Ait}$, and hygroscopicities $\kappa_{acc}$, $\kappa_{Ait}$), and of the updraft velocity $w$ for $N_d$. In previous model sensitivity studies of monomodal ASDs, aerosol - and updraft -limited regimes were defined in which $N_d$ depends linearly on $N_a$ or $w$ (Reutter et al., 2009). Using this concept, we show that Aitken mode particles are not activated if updraft-limited conditions prevail in the presence of a dominant accumulation mode (high $N_{a,acc}$). Also aerosol-limited conditions do not occur as by far not all Aitken mode particles are activated (for w $\leq$ 3 ms$^{-1}$, i.e., updraft velocities of abundant cloud types) and thus the transitional regime (between $Na$ and $w$ limitations) exists over wider parameter spaces than in the presence monomodal accumulation mode ASDs. When ASDs are dominated by an Aitken mode, we find that $N_d$ is highly sensitive to $w$ even at low $N_{a,Ait}$ which implies that the $w/N_a$ regime limits as identified previously for accumulation mode ASDs are not always applicable, but they depend on parameter value combinations of $D_g$, $N_a$, $\kappa$ and $w$.

Exceeding the previous framework that was restricted to $w$ and $N_a$-limitations, we show that the sensitivities of $N_d$ in the transitional and $w$-limited regimes equally depend on $w$ and $\kappa$. Therefore, we explored in detail the sensitivity of $N_d$ to $\kappa$, $\xi(\kappa)$, as a function of $w$ for ASDs that differ in the number of modes (mono- or bimodal), $N_{a,Ait}/N_{a,acc}$ and total $N_a$. Based on the patterns of $\xi(\kappa)$ as a function of $w$ and $\kappa$, we analyse the dependence of the $w,\kappa$ range, above which Aitken mode particles contribute to $N_d$ on the ASD parameters. We show that $\xi(\kappa)$ exhibits minimum values for $w,\kappa$ combinations for which the smallest activated particle size ($D_{min}$) is near the Hoppel minimum and increases when smaller Aitken mode particles are activated. Defining lines near these $w,\kappa$ combinations as the minimum threshold , it can be estimated under which aerosol ($N_a,\kappa,D_g$) and $w$ conditions Aitken mode particles start contributing to $N_d$. We conclude that the most important requirements are a low number concentration of total and accumulation mode particles ($N_a$, $N_{a,acc}$) and/or a large mode diameter of the Aitken mode ($D_{g,Ait}$). While this ranking repeats previous findings for sensitivities to monomodal ASDs (e.g., Ervens et al. (2005); Reutter et al. (2009); Ward et al. (2010); Cecchini et al. (2017); Pardo et al. (2019)), our analysis exceeds these studies as it evaluates the relative importance of these parameters of accumulation and Aitken modes for the activation of Aitken mode particles.

Applying this framework to typical ambient aerosol conditions, it seems likely that above the ocean where aerosol loading is usually low and ASDs often exhibit bimodal shapes with very hygroscopic particles (Wex et al., 2016; Braga et al., 2021), Aitken mode particles are activated to cloud droplets. This confirms findings in marine stratocumulus clouds with moderate $w \geq 0.5$ m s$^{-1}$ (Schulze et al., 2020). Given that marine stratocumuli comprise a large fraction of global cloud coverage, the contribution of Aitken mode particles to $N_d$ above the ocean should thus be included in global estimates of aerosol-cloud inter-actions. Similarly, our concept is consistent with the large observed fractions of activated Aitken mode particles at Arctic sites (Komppula et al., 2005). Contrary, it implies that in highly polluted regions even high $N_{a,Ait}$ (e.g., in megacities, Mönkkönen et al. (2005)) are not relevant in stratocumulus and shallow cumulus clouds as droplets will only form on accumulation mode particles.

Global model studies have identified large uncertainties in CCN number concentration and $N_d$ predictions due to the assumptions associated with Aitken mode particle properties, specifically in the Southeast US, Europe and to a small extent in the Amazon region (Lee et al., 2013; Chang et al., 2021). Our framework, together with global maps of $\kappa_{Ait}$ and $\kappa_{acc}$ (e.g.,





Chang et al. (2017)), will help reducing these uncertainties and constraining aerosol-cloud interactions in regions where Aitken

mode particles affect cloud properties.



## Appendix A

**Table A1.** Definition of parameters that are used in the discussion

| Parameter | Description |
|---|---|
| $D_g$ | Geometric mean mode diameter |
| $D_{g,Ait}$ | Mean Aitken mode diameter |
| $D_{g,acc}$ | Mean Aitken mode diameter |
| $D_{min}$ | D of smallest particle that forms a cloud droplet |
| D | Diameter of dry particle |
| $F_{act,Ait}$ | Activated fraction of Aitken mode particles $N_{d,Ait}/N_{a,Ait}$ |
| $F_{act,acc}$ | Activated fraction of Aitken mode particles $N_{d,acc}/N_{a,acc}$ |
| $\kappa$ | Hygroscopicity parameter |
| $\kappa_{Ait}$ | Hygroscopicity parameter for Aitken mode particles |
| $\kappa_{acc}$ | Hygroscopicity parameter for accumulation mode particles |
| $N_a$ | Particle number concentration |
| $N_{a,acc}$ | Particle number concentration of Aitken mode particles |
| $N_{a,Ait}$ | Particle number concentration of accumulation mode particles |
| $N_d$ | Predicted droplet number concentration |
| $N_{d,Ait}$ | Number concentration of droplets formed on Aitken mode particles |
| $N_{d,acc}$ | Number concentration of droplets formed on accumulation mode particles |
| s | Saturation |
| $s_{eq}$ | Equilibrium saturation based on Köhler theory |
| $S_{max}$ | supersaturation |
| $\sigma_g$ | Geometric standard deviation for mode |
| w | Updraft velocity |
| $\xi(\kappa)$ | Sensitivity of $N_d$ to $\kappa$ (Equation E.4) |
| $\xi(N_a)$ | Sensitivity of $N_d$ to $N_a$ (Equation E.5) |


*Code and data availability.* Details on the model codes and further model results can be obtained from the corresponding authors upon request.

*Author contributions.* MLP and BE led the study and wrote the manuscript with input from all coauthors. MZ and BE performed the model
simulations. MZ, RCB, OOK, and UP commented on the manuscript.

*Competing interests.* The authors declare that they have no conflict of interest.

*Acknowledgements.* This work has been supported by the French National Research Agency (ANR) (grant no. ANR-17-MPGA- 0013) and the Max Planck Society (MPG).





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
