# Peer review of "Aitken mode particles as CCN in aerosol- and updraft-sensitive regimes of cloud droplet formation"

_Atmospheric Chemistry and Physics, 2021_

## Referee Comment (RC2)

Review Report on "Aitken mode particles as CCN in aerosol- and updraft-sensitive regimes of cloud droplet formation" by Pöhlker et al

Jingyi Chen

**General Comments:**

It is widely thought that aerosols in accumulation mode contribute to the activation processes, and aerosols in Aitken mode are usually ignored in most studies. This manuscript demonstrates the role of aerosols in Aitken mode by adiabatic cloud parcel model. They found that the activation of Aitken mode plays roles in the ACI, especially in the dependence of Nd to hygroscopicity $\xi(\kappa)$. They also show the regime patter of Nd and $\xi(\kappa)$ in many sensitivity studies with monomodal and bimodal. Overall, this manuscript adds value in the community that it provides a theoretical analysis of the roles of Aitken model aerosols in the activation processes, which is necessary but hasn't been done before. However, some details need to be revised or clarified. I recommend a minor revision.

**Specific comments:**

1.  It is not surprising that at higher w and higher $\kappa$ that Aitken mode is more important, because higher values of either one or both favor the activation of smaller particles. It is good that authors include threshold to tell if Aitken mode is important ($F_{Ait}>0.05$). It is better to also show which part of Aitken mode is important, because $F_{Ait}$ is smaller than $F_{accu}$ in the simulations shown in the manuscript.

2.  Fixed values for droplet criterion lead to a problem that those particles will grow to the cloud droplet at high altitudes but are not taken into account as droplets close to cloud base, because the time that particles expose to the supersaturation is not long enough. That is why $\xi(\kappa)$ increases (Figure 2f) and coincides with the orange box in Figure S4. In other saying, the increasing $\xi(\kappa)$ above $S_{max}$ might only be a manifestation from the assumption of fixed values for droplet criterion. I suggest more discussions on that with more similar figures as Figure 2e-f, using the droplet criterion in Reutter et al 2009.

3.  Figure 3 is presented at about 20m above $S_{max}$, where $\xi(\kappa)$ has not reached to steady state based on Figure 2f. When the parcel further goes higher, those cases with high $\kappa$

will correspond to the higher $\xi(\kappa)$. I guess that at 40m, the minimum of $\xi(\kappa)$ in the w-$\kappa$ will move to the smaller w and $\kappa$ side. Those differences might not exist if not using the fixed criterion for $N_d$. I suggest two more figures 1) a similar figure as Figure 3 except at 40m with same fixed size as criterion for $N_d$, and 2) a similar figure as Figure 3 but using the same criterion as Reutter et al 2009.

4. Figure S3 shows the vertical profiles of supersaturation and is only mentioned briefly on Line 144. Activated of Aitken mode aerosols depends on the supersaturation and also feedbacks to the supersaturation through the second term in Equation E2. In Fugure S3, it is interesting to see at 1m/s, adding Aitken mode suppresses the supersaturation and decreases the cloud base height, while at 0.2 m/s, adding Aitken mode enhances the supersaturation and increases the cloud base height. More discussion on that will be a good addition to this manuscript.

5. Line 209-211, with different combination of w and $\kappa$, same $D_{min}$ and $\xi(\kappa)$ can be reached, as where the lines with different color cross in figure 3b and 3c. How about those points with same $D_{min}$ but different $\xi(\kappa)$? In other saying, with a same aerosol size distribution and same $D_{min}$, why $\xi(\kappa)$ are different?

6. In Figure S6b, the minimum points of the green and cyan lines are not at the same size of Hoppel minimum diameter. Why is that?

7. Line 234, it is easy to understand that high $\xi(N_a)$ in aerosol-limited regime, but why the $\xi(N_a)$ is also large in the w-limited regime? In w-limited regime, $N_d$ is supposed to be dependent on w and much less dependent with $N_a$, thus $\xi(N_a)$ should be any values close to 0. Also, I don't understand why there is a minimum in Figure 4b.

8. First of all, the color scheme in Figure 4c is very hard to read. Does the more transparent color scheme represent the total $N_d$ or only the $N_d$ corresponding to aerosols in accumulation mode? Secondly, the authors stated that the differences between simulation ASD I and ASD III represent the effects of Aitken mode. However, the total aerosol number concentration is different in these two simulations. In my opinion, $N_a$-limited regime corresponds to the scenario that $N_a$ is small but doubling $N_a$ itself contributes to the vanishing of aerosol-limited regime. In this case, the differences

between ASD I and ASD III include both change in total Na and the number of modal. I have same comments for Line 258-259. In other saying, the vanishing of aerosol-limited regime might not be a feature for monomodal v.s. bimodal. It might be a feature due to the increasing $N_a$.

9. Those contour figures showing the dependence of $\xi(\kappa)$ on w and $\kappa$ are very noisy, especially V.c in Figure 5 (right bottom one). There is no way to tell the trends in V.c in Figure 5. My experience tells me that increasing the number of particle bins (i.e., 545 particle size classes mentioned in section 2.2.1) can improve the noisy problems. Hope this also helps to improve the quality of that figure. Moreover, in Figure 2f, it is hard to tell the reverse relationships between $\xi(\kappa)$ and $\kappa$ at high altitudes. Perhaps less lines can show the reverse better.

10. Line 305 "Nd is highly sensitive to $\kappa$ and w, and less to $N_{a,Ait}$". The activation fraction in Aitken mode is much smaller than that in the accumulation mode. That might be caused by the small lower size limit of Aitken mode in those simulations. What will happen if the simulation only includes the relative larger particles in Aitken mode so that the activation fraction in Aitken mode is comparable to that in accumulation mode? Na might be important in that way. Section 3.3.2 shows the impacts of Dg on the $\xi(\kappa)$. I think it is better to have more discussion about relative importance of $N_a$, w and $\kappa$ when $D_{g,Ait}$ is larger. This is also related to my first comment that it will be useful to provide the information that which part of Aitken mode is more important for ACI.

**Minor comments:**

1. the section 3.1. presents in the way that w and $\kappa$ are equally important, it is better to move at least one between Figure S1 and S2 (better both) to the main text to show the dependence on w.

2. Line 10, add a comma after $N_d$.

3. Line 175, change "S3" to "S4".

4. Line 257, impacts of $D_g$ has not been shown here, so it is early to include it in the summary here.

---

## Author Comment (AC1)

**Author response to referee comments**

We thank both referees for their positive evaluation of our manuscript and the constructive suggestions for improvement. We respond to all comments in detail below. Referee comments are in black, our responses in blue and manuscript text in *italic* and new text in *red*. Line numbers refer to the revised manuscript..

**Referee 1**

This manuscript investigates the contribution of Aitken mode particles to cloud droplet formation, and the sensitivity of this contribution to the main influencing factors. The study is based on a large set of simulations made using an adiabatic cloud parcel model. While the used approach itself it by no means novel, the simulations conducted here and their interpretation clearly adds new insight into the topic of cloud droplet activation. I therefore consider this paper original enough for publication. The conducted study is scientifically sound, and there no apparent errors in methods or interpretation of results. I recommend accepting this paper for publication after the authors have addressed the few comments outlined below.

Main issues

Referee Comment 1: Discussion of the results of simulations is quite detailed and requires, in many places, a lot of attention from a reader. While I accept this feature in general, there is one specific place that need to be modified: Figure 3. This figure (especially panes b and c ) is way too complicated, with multiple axises and legends that are difficult to digest. I would strongly recommend simplifying this figure, or even splitting it into 2 parts. The text discussing this figure might also be worth simplification.

Author response: We thank the referee for pointing out the issues with Figure 3. We decided not to split this figure into two since the comparison of the $D_{min}$ is the main message of this figure. Instead, we revised the figure and added a new panel c that shows now the $D_r$ange for the two sets of simulations, i.e. with constant $\kappa$ for the full range of $w$ and vice versa. We replaced Figure 3 in the main manuscript and also Figures S5 and S6 in the supplement. In the latter two figures, we omitted the vertical dashed lines for clarity.

We also revised the text describing these figures accordingly (l. 204ff):

*Parallel to the axes, six lines are marked for three $\kappa$ values (vertical lines at $\kappa = 0.7$  (orange), 0.3  (blue), and 0.04  (red)), and three updraft velocities (horizontal lines at $w = 2.9$ m s$^{-1}$  (orange), 1.0 m s$^{-1}$  (blue) and 0.2 m s$^{-1}$  (red)).*

*     Each $\xi(\kappa)$ can be related to a $D_{min}$ (Figure 2 in the manuscript). This relationship is shown in Figure R1-1b, where the $\xi(\kappa)$ values along the vertical lines are overlaid by the aerosol size distribution.*

* Figure 3c shows the $D_{min}$ range that is covered by the simulations for the three constant $\kappa$ values. The end points of the $D_{min}$ ranges in Figure 3b and c are connected by vertical dashed lines. In the same way, Figure 3d shows $D_{min}$ for $\xi(\kappa)$ i.e. along the horizontal lines in panel a   blue and orange bars in the lower portion of Figure 3c .*

[Figure]

Figure R1-1: (Figure 3 in revised manuscript): a) Sensitivity of cloud droplet number concentration to aerosol hygroscopicity ($\xi(\kappa)$) as a function of $\kappa$ and $w$ for ASD III.b at 20 m above $S_{max}$. b): $\xi(\kappa)$ (right axis) as a function of $D_{min}$ for $\kappa = 0.04$ (red), $\kappa = 0.3$ (blue), and $\kappa = 0.7$ (orange). c) Ranges of $D_{min}$ for the simulations in panel b) and d). b): $\xi(\kappa)$ (right axis) as a function of $D_{min}$ for $w = 0.2$ m s$^{-1}$ (red), $w = 1.0$ m s$^{-1}$ (blue), and $w = 2.9$ m s$^{-1}$ (orange).

[Figure]

Figure R1-2: (Figure S5 in revised manuscript): Same as Figure R1-2 but for $N_a = 200$ cm$^{-3}$

[Figure]

Figure R1-3: (Figure S6 in revised manuscript):Same as Figure R1-2 but for $N_a = 5000$ cm$^{-3}$

Referee Comment 2: Mathematically, two modes in a particle number size distribution overlap each other because a log-normal mode a tail that continues for infinity. In the cases simulated in this work, the overlapping region is a notable fraction of the overall particle population, as the two modes are centered relatively close to each other. As a result, some Aitken mode particles are always larger (and thereby activate easier to cloud droplets) than some accumulation mode particles. In reality, this mathematical feature might be acceptable if the two modes represented different sources and thereby had potentially very different chemical composition. But this feature is highly questionable in aged air masses, like in cloud-processed air where all particle to the right of the Hoppel minimum should be counted as accumulation mode particle and those left to it as Aitken mode particles. The authors should bring up this issue and discuss it shortly in the paper. My main concern here is that does this upper tail of the Aitken mode (or the part of the tail that in reality should be called as accumulation mode particles) influence notably the estimated contribution of Aitken mode particles the cloud droplet population (this might be important as the criterion for notable contribution here is that 5% of cloud droplets originate from the Aitken mode, see Figure 7).

Author response: We agree with the referee that different criteria are commonly be applied to distinguish Aitken and accumulation mode. These criteria are either based on size (larger or smaller than the Hoppel minimum) or based on sources that lead to an overlap of the 'tails' of two modes. We realized that our Figure 1 was misleading in this regard as it implied that we used in our models overlapping tails of the two modes. However, in the model we used the Hoppel minimum as the threshold to distinguish between the two modes. Therefore, we can clearly define the activated fractions of $F_{Ait}$, which corresponds for example for $F_{Ait} = 0.05$ to particles in the size range of 66 - 73 nm, i.e. smaller than the Hoppel minimum. To avoid this misunderstanding, we revised Figure 1

[Figure]

Figure R1-4: Cumulative number fraction of Aitken mode particles. Size thresholds are indicated for $F_{Ait}$ = 0.05 and 0.5

[Figure]

Figure R1-5: Schematic of model input aerosol size distributions (ASDs, designated I-V, a-c) with different number concentrations of Aitken mode particles ($N_{a,Ait}$, red) and accumulation mode particles ($N_{a,acc}$, blue). Particle number concentrations are given in units of $cm^{-3}$. The modal geometric mean diameters are $D_{g,Ait} = 0.037\,\mu m$ and $D_{g,acc} = 0.145\,\mu m$ .

in the manuscript (Figure R1-5 in this response) and
also clarified this in the manuscript (l. 115ff):

>   *We distinguish the two modes by the diameter of the Hoppel minimum ($\sim$0.07 $\mu m$). While strictly both modes have 'tails' beyond the Hoppel minimum (Aitken mode particles being larger and accumulation mode particles being smaller than the Hoppel minimum), we do not consider this in our model. This simplification seems justified since the mode classification in measured ASDs are usually ascribed based on particle size and not based composition which may differ because of different sources of particles in the two modes.*

**Minor issues:**

Referee Comment 3: line 188: It can be concluded that...

Author response: We thank the referee for noticing the omission. We added 'that'.

Referee Comment 4: Figure 7: The labels of the panels (a, b, c and d) are missing from this figure.

Author response: We added the labels a - d to the panels of Figure 7.

**Referee 2 (Jingyi Chen)**

**General Comments:**

It is widely thought that aerosols in accumulation mode contribute to the activation processes, and aerosols in Aitken mode are usually ignored in most studies. This manuscript demonstrates the role of aerosols in Aitken mode by adiabatic cloud parcel model. They found that the activation of Aitken mode plays roles in the ACI, especially in the dependence of Nd to hygroscopicity $\xi(\kappa)$. They also show the regime patter of Nd and $\xi(\kappa)$ in many sensitivity studies with monomodal and bimodal. Overall, this manuscript adds value in the community that it provides a theoretical analysis of the roles of Aitken model aerosols in the activation processes, which is necessary but hasn't been done before. However, some details need to be revised or clarified. I recommend a minor revision

**Specific comments:**

Referee Comment 1: It is not surprising that at higher w and higher $\kappa$ that Aitken mode is more important, because higher values of either one or both favor the activation of smaller particles. It is good that authors include threshold to tell if Aitken mode is important (FAit >0.05). It is better to also show which part of Aitken mode is important, because FAit is smaller than Faccu in the simulations shown in the manuscript.

Author response: We thank the reviewer for this suggestion. As we assume internally mixed aerosol, i.e. all particles have the same composition ($\kappa$).Therefore, the largest Aitken mode particles are activated first. A fraction of $F_{Ait}$ corresponds to all Aitken mode particles that are larger than $\sim$66 nm; $F_{Ait} = 0.5$ corresponds to particles larger than $\sim$38 nm (Figure R2-1). In most simulations, all accumulation mode particles are activated before any Aitken mode particles as for a given $\kappa$ and $w$ only particle size determines their activation. The only exception are the simulations presented in Section 3.3.1 in which $F_{acc} < 1$ and $F_{Ait} \leq 0.2$ , depending on $\kappa_{acc}$ (Figure S9).

[Figure]

Figure R2-1: Cumulative number fraction of Aitken mode particles. Size thresholds are indicated for $F_{Ait}$ = 0.05 and 0.5

Referee Comment 2: Fixed values for droplet criterion lead to a problem that those particles will grow to the cloud droplet at high altitudes but are not taken into account as droplets close to cloud base, because the time that particles expose to the supersaturation is not long enough. That is why $\xi(\kappa)$ increases (Figure 2f) and coincides with the orange box in Figure S4. In other saying, the increasing $\xi(\kappa)$ above Smax might only be a manifestation from the assumption of fixed values for droplet criterion. I suggest more discussions on that with more similar figures as Figure 2e-f, using the droplet criterion in Reutter et al 2009.

Author response: Reutter et al. (2009) defined droplets as those particles for which their critical supersaturation ($S_{crit}$) is exceeded. In such simulations, the drop number concentration $N_d$ only increases until the maximum supersaturation is reached. Above this threshold, the supersaturation decreases and therefore, $N_d$ stays constant. This height at which $S_{max}$ is reached depends on the hygroscopicity of the particles ($\kappa$); it is indicated in Figure 2 as the black line across the colored lines.

We do not think that a fixed size threshold represents a 'problem'. We have chosen this drop definition (as opposed to the one based on $S_{crit}$) as any conclusions are more applicable and comparable to measurements. The critical diameter is based on the theory of droplet activation (Köhler theory). However, cloud probes cannot measure directly supersaturation in clouds and therefore cannot quantify the number of particles for which the critical supersaturation has been exceeded. Instead, they apply a size threshold above which particles are counted as droplets. Since droplet size is an essential parameter to determine cloud radiative forcing, we think

that using a instrumental size threshold is more useful in our studies that should guide observationally-based model studies rather than using the threshold as defined based on $S_{crit}$. We added in line 189 ff:

*Since $N_d$ would not change above $S_{max}$, also $\xi(\kappa)$ would be constant above this level.*

Referee Comment 3: Figure 3 is presented at about 20m above Smax, where $\xi(\kappa)$ has not reached to steady state based on Figure 2f. When the parcel further goes higher, those cases with high $\kappa$ will correspond to the higher $\xi(\kappa)$. I guess that at 40m, the minimum of $\xi(\kappa)$ in the w-$\kappa$ will move to the smaller w and $\kappa$ side. Those differences might not exist if not using the fixed criterion for Nd. I suggest two more figures 1) a similar figure as Figure 3 except at 40m with same fixed size as criterion for Nd, and 2) a similar figure as Figure 3 but using the same criterion as Reutter et al 2009.

Author response: Figure 3 shows results at 20 m above the levels of $S_{max}$. This height corresponds to $\sim 35$ m above the level where RH = 100% in Figure 2e and f. We realized that referring to two different heights might have caused confusion. Therefore we added in line 202ff:

*This height corresponds approximately to 35 m above cloud base, i.e. the height at which RH = 100% (Figure 2).*

Referee Comment 4: Figure S3 shows the vertical profiles of supersaturation and is only mentioned briefly on Line 144. Activated of Aitken mode aerosols depends on the supersaturation and also feedbacks to the supersaturation through the second term in Equation E2. In Fugure S3, it is interesting to see at 1m/s, adding Aitken mode suppresses the supersaturation and decreases the cloud base height, while at 0.2 m/s, adding Aitken mode enhances the supersaturation and increases the cloud base height. More discussion on that will be a good addition to this manuscript.

Author response: We thank the referee for pointing out these trends, and we apologize for the confusion. We realized that Figure S2d and S2f were mixed up. We replaced the figure by the correct one. Since now the figure panels for ASDI.b and III.b at a given updraft velocity are nearly identical, we did not modify or add any text.

Referee Comment 5: Line 209-211, with different combination of w and $\kappa$, same Dmin and $\xi(\kappa)$ can be reached, as where the lines with different color cross in figure 3b and 3c. How about those points with same Dmin but different $\xi(\kappa)$? In other saying, with a same aerosol size distribution and same Dmin, why $\xi(\kappa)$ are different?

Author response: We revised and clarified Figure 3 in response to comments by Referee 1. In brief, the slope of the ASD near $D_{min}$ determines the change in $N_d$ and therefore $\xi(\kappa)$, which is defined as the change in $N_d$ with a change in $\kappa$. As the referee points out correctly, different combinations of $w$ and $\kappa$ can result in the same $D_{min}$. Depending on the values of condensation and updraft terms, the required changes in $\kappa$ to yield the same $D_{min}$ are different. To yield the same change in $D_{min}$ (and therefore an equal change in $N_d$) the required change in $\kappa$ depends on the value of the updraft term. Thus, for different $w$, the same change in $\kappa$ can result in different $\Delta N_d$ and also in $\xi(\kappa)$. This is also reflected in the panels c in Figures 3, S5 and S6 that imply different sensitivities of $D_{min}$ to $\kappa$ and $w$ depending on $N_a$ which also controls the condensation term.

Referee Comment 6: In Figure S6b, the minimum points of the green and cyan lines are not at the same size of Hoppel minimum diameter. Why is that?

Author response: The referee is correct that the minima in the $\xi$ lines do not always match up with the size range of the Hoppel minimum. This effect is most pronounced in Figure S6, i.e at the highest $N_a$, whereas the minima of the $\xi(\kappa)$ curves for the the lowest $N_a$ seems to exactly line up with the Hoppel minimum (Figure S5). However, even at $N_a = 1000$ cm$^{-3}$, the minima of the $\xi(\kappa)$ curves are sightly shifted to larger sizes. The higher $N_a$, the lower is the supersaturation in cloud (Figure S7). At such low supersaturation, an increase in $w$ might not significantly enhance $N_d$ as additionally activated particles decrease the supersaturation, which is relatively low due to the dominating condensation term. Such 'buffering effects' result in nearly constant $N_d$ despite higher $w$ or $\kappa$. Such conditions were termed 'updraft-limited regime' by Reutter et al (2009), since $N_d$

mainly depends on $w$ ad not on $N_a$. Only if the updraft significantly increases, more particles can be activated which is reflected by the increasing $\xi(\kappa)$ at high $w$ in Figure S6b. We added it to the discussion of Figres S5 and S6 (l. 228ff):

*While for $N_a = 200$ cm$^{-3}$ and 1000 cm$^{-3}$ the minima in the $\xi(\kappa)$ curves coincide with the Hoppel minimum of the ASD, the minimum of $\xi(\kappa)$ is shifted to somewhat larger sizes for $N_a = 5000$ cm$^{-3}$ (Figure S6). At such high $N_a$, the supersaturation is very low (Figure S7). Under these conditions, an increase in $w$ or $\kappa$ might result in only small changes in $N_d$ because of buffering effects, i.e. the activation and efficient growth of additional droplets suppresses the supersaturation and prevents further activation.*

Referee Comment 7:

a) Line 234, it is easy to understand that high $\xi(Na)$ in aerosol-limited regime, but why the $\xi(Na)$ is also large in the w-limited regime? In w-limited regime, Nd is supposed to be dependent on w and much less dependent with Na, thus $\xi(Na)$ should be any values close to 0.

Author response: $\xi(Na)$ is defined as a change in $N_d$ with a change in $N_a$. The referee is correct that in the aerosol-limited regime $\xi(Na) \sim 1$ because $N_d$ can only increase if $N_a$ increases (upper left corner of Figure 4b). At high $N_a$ and low $w$ (bottom right corner of Figure 4b), $N_d$ is largely determined by $w$ (or $\kappa$), and largely independent of $N_a$. Thus, a change in $N_a$ that results in the same $N_d$ leads to different values of $\xi(Na)$. These values are larger for larger absolute changes in $N_a$ (Eq.-E5) which is reflected in the highest values of $N_a$ at the highest $N_a$.

b) Also, I don't understand why there is a minimum in Figure 4b. Author response: The minimum in the $\xi(Na)$ pattern coincides approximately with the line for F = 0.7. For this activated fraction the slope of the cumulative ASD is highest. It means that any activation beyond this fraction leads to a relatively smaller increase in $N_d$. We added to the manuscript (l. 253ff):

*The $\xi(N_a)$ pattern in Figure 4b exhibits a minimum when the activated fraction is $\sim 0.7$. This fraction corresponds to the size range at which the cumulative ASD has the largest slope. If more particles are activated, the relative change in $N_d$ and therefore in $\xi(Na)$ become smaller.*

Referee Comment 8:

a) First of all, the color scheme in Figure 4c is very hard to read. Does the more transparent color scheme represent the total Nd or only the Nd corresponding to aerosols in accumulation mode?

Author response: In the upper panels (a, c, e) in the original Figure 4, the two different color schemes represented the values for the individual modes ($N_{d,acc}$ and $N_{d,Ait}$). We revised the figure by (1) enlarging the color bars and (2) replacing the scale for the Aitken mode by a black/white color scale (Figure R2-2). We also realized that the figure caption was incomplete and we added the information that all simulations were performed for a single $\kappa = 0.7$

b) Secondly, the authors stated that the differences between simulation ASD I and ASD III represent the effects of Aitken mode. However, the total aerosol number concentration is different in these two simulations. In my opinion, Nalimited regime corresponds to the scenario that Na is small but doubling Na itself contributes to the vanishing of aerosol-limited regime. In this case, the differences between ASD I and ASD III include both change in total Na and the number of modal. I have same comments for Line 258-259. In other saying, the vanishing of aerosol-limited regime might not be a feature for monomodal v.s. bimodal. It might be a feature due to the increasing Na.

Author response: The referee is correct that doubling $N_a$ would lead to a shift in the regimes in the w-$N_s$space as indicated in Figure R2-2 (= Figure 4 in the manuscript), i.e. from aerosol-limited to transitional if a mono-modal ASD is considered, in agreement with the findings by Reutter et al. (2009). However, the situation is different

[Figure]

Figure R2-2: (Figure 4 in revised manuscript): Upper panels: $N_d$ as a function of $w$ and $N_a$ for ASD and $\kappa = 0.7$. a) I, c) III and e) V. Color scale: $N_{d,acc}$, black/white scale: $N_{d,Ait}$. Bottom panels: Corresponding $\xi(N_a)$ for ASD b) I, d) III and f) V. Solid black lines show contours of activated fraction of the accumulation mode ($F_{act,acc}$), dashed grey lines show contours of activated fraction of the Aitken mode ($F_{act,Ait}$).

in our simulations as we do not scale up the ASD by doubling $N_a$ but we add a second mode with different particle sizes. Under conditions when Aitken mode particles are not activated, the results of the simulations with ASD I and ASD III are identical. Thus, Figure R2-2d corresponds to the left half ($N_a \leq 2500$ cm[-3] of Figure R2-2b. To clarify this, we had added the second x-axis to Figure R2-2d. To further clarify it, we added to the text (l. 257ff):

*Figure 4d shows the same $\xi(N_a)$ patterns as the part of Figure 4b for $N_a \leq 2500$ cm[-3]. Under such conditions, only accumulation mode particles are activated and thus contribute to $N_d$ and to $\xi(N_a)$.*

Referee Comment 9:

a) Those contour figures showing the dependence of $\xi(\kappa)$ on w and $\kappa$ are very noisy, especially V.c in Figure 5 (right bottom one). There is no way to tell the trends in V.c in Figure 5.

My experience tells me that increasing the number of particle bins (i.e., 545 particle size classes mentioned in section 2.2.1) can improve the noisy problems. Hope this also helps to improve the quality of that figure.

textcolorRoyalBlueAuthor response: All figure panels in Figures 5 show the results for 810 simulations (30 $w$ and 27 $\kappa$ values). We performed simulations with higher resolution of these values but the contour plots look very similar. In Figure R2-3, we show an example of results for 2180 size classes (i.e. four ties more than in the original simulations).

As the patterns do not significantly change, we decided not to update the figures in the revised manuscript. We would like to emphasize that the main purpose of Figure 5 is to show the trends in the location of the $\xi(\kappa)$ minimum as a function of ASD shape and N$a$; we do not discuss the absolute values of $\xi(\kappa)$.

[Figure]

Figure R2-3: Comparison of model results for ASD V: Upper row: ASD V.a ($N_a = 100$ cm$^{-3}$ a) original figure (545 size classes, 27 $\kappa$ values 30 $w$ values, b) 2180 size classes, 27 $\kappa$ values, 30 $w$ values, Bottom row: ASD V.c ($N_a = 2500$ cm$^{-3}$; c) 545 size classes, 53 $\kappa$ values, 59 $w$ values

Author response: We repeated the simulations for Figure 5 V.c with 2180 size classes, i.e increased by a factor of 4 as compared to the original simulations. We also repeated them for 53 $\kappa$ and 59 $w$ values to also enhance this resolution by a factor of four while keeping the number of size classes at 545. The results are shown in Figure R2-3. We realize that the features and absolute numbers slightly change within the figures, depending on the resolution of size classes or $\kappa$ and $w$ values. However, the overall message does not change: We show in the figure that the $\xi(\kappa)$ patterns for the high $N_a$ simulations (bottom row of Figure 5) maybe considered a continuation of the patterns obtained from the low $N_a$ simulations. For example, the 'red streaks' for high $\xi(\kappa)$ values that are very prominent in Figure R2-3f are only seen at the bottom of Figure R2-3a. Similarly, as we had pointed out already in the manuscript, the features for ASD V can be considered a continuation of the patterns of ASD I to smaller $\kappa$ values. These trends demonstrate that different combinations of the parameters in the condensation and updraft terms ($D_g$, $N_a$, $w$, $\kappa$) could result in the same $\xi(\kappa)$ values.

b) Moreover, in Figure 2f, it is hard to tell the reverse relationships between $\xi(\kappa)$ and $\kappa$ at high altitudes. Perhaps less lines can show the reverse better.

Author response: We followed the referee's suggestion removed ~50% of the lines in the original Figure 2f. As we do not see any significant difference in the clarity of the figure, we decided to leave the original figure in the manuscript. Again, we would like to point out that we do not discuss the exact $\xi(\kappa)$ values but we want to show trends caused by the presence of an Aitken mode. This message is clearly seen in Figure 2f, i.e., that the $\xi(\kappa)$ for ASD III.B cover a much narrower range than those for ASD I.B and that the order of lines (i.e different $\kappa$ values) is different for $\xi(\kappa)$ than for $D_{min}$ and $N_d$.

[Figure]

Figure R2-4: Identical to Figure 2f in the original manuscript except that fewer lines for different $\kappa$ values are shown.

Referee Comment 10: Line 305 "Nd is highly sensitive to $\kappa$ and w, and less to Na,Ait". The activation fraction in Aitken mode is much smaller than that in the accumulation mode. That might be caused by the small lower size limit of Aitken mode in those simulations. What will happen if the simulation only includes the relative larger particles in Aitken mode so that the activation fraction in Aitken mode is comparable to that in accumulation mode? Na might be important in that way. Section 3.3.2 shows the impacts of Dg on the $\xi(\kappa)$). I think it is better to have more discussion about relative importance of Na, w and $\xi(\kappa)$ when Dg,Ait is larger. This is also related to my first comment that it will be useful to provide the information that which part of Aitken mode is more important for ACI.

Author response: The referee is correct that the importance of the Aitken mode for aerosol-cloud interactions increases if its activated fraction increases. However, under typical conditions, there is rarely any case under which ASDs only exhibit an Aitken mode and the accumulation mode is completely absent. If an accumulation mode is present, these particles will mostly determine the supersaturation in cloud and therefore also control the fraction of the Aitken mode that can be activated. The activated fraction of the accumulation mode will generally be higher than that of the Aitken mode as accumulation mode particles are (1) larger and (2) usually more hygroscopic than Aitken mode particles.

Figures S9 shows that under common conditions, it does not seem realistic to have equal activated fractions of both modes. Even if $\kappa_{acc} = 0.1$ and $\kappa_{Ait} = 1$, $F_{Ait}$ does not exceed ~0.3 whereas $F_{acc}$ ~1 (upper right corner of Figure S9a). Generally, we show in Figure S9 that at most 30% of the Aitken mode is important for the conditions chosen here $\kappa \leq 1$; $w \leq 3$ m s$^{-1}$), which means that the largest 30% of the Aitken mode particles are activated (i.e. all Aitken mode particles with diameters greater than ~45 nm, Fig. R2-1).

We are not sure about the relevance of a comparison of situations where $F_{acc} \sim F_{Ait}$. If the referee could point us to any observational evidence that such conditions are encountered in the atmosphere, we will be happy to discuss them in the manuscript. Given that the usual conditions rather seem to point to situations of $F_{acc} > F_{Ait}$, we did not add any references or discussion to the existing text in the introduction or Section 3.3.1. In order to clarify that only the largest Aitken mode particles are activated, we added to the manuscript (l. 346ff):

*The large overlap of the activated fractions from both modes in Figure S8 a ($\kappa_{acc} = 0.1$) demonstrates that at w ~1 ms$^{-1}$, large Aitken mode particles with $\kappa_{Ait} \sim 0.3$ may grow to droplet sizes even though only ~70% of accumulation mode particles are activated whereas the smallest ~30% of the accumulation mode particles have not been activated yet.*

**Minor comments:**

Referee Comment 11: the section 3.1. presents in the way that w and $\kappa$ are equally important, it is better to move at least one between Figure S1 and S2 (better both) to the main text to show the dependence on w.

Author response: We thank the referee for this suggestion. However, in order to keep the number of figures reasonable and to emphasize the focus of the paper, which is the effect of Aitken mode particles on $N_d$, we decided not to move Figures S1 and/or S2 to the main manuscript. These figures show conditions under which only accumulation mode particles are activated. The sensitivity of $w$ to $N_d$ for accumulation mode particles has been discussed in many previous studies (e.g. Ervens et al., 2005; Reutter et al., 2009). Therefore, we think that these results are not essential for our conclusions to be shown in the main part of the manuscript.

Referee Comment 12: Line 10, add a comma after Nd".
Author response: The comma was added.

Referee Comment 13: Line 175, change "S3" to "S4".
Author response: We corrected the typo.

Referee Comment 14: Line 257, impacts of Dg has not been shown here, so it is early to include it in the summary here.
Author response: The referee is correct that we did not discuss the sensitivity to different $D_g$ for accumulation or Aitken mode yet. However, the comparison of ASD I and ASD V in Figure 4 represents a comparison of the influence of different $D_g$, i.e. $D_{g,Ait} = 0.037$ $\mu$m and $D_{g,acc} = 0.145$ $\mu$m. Therefore, we did not change any text at this place.